# dTRPO: Trajectory Reduction in Policy Optimization of Diffusion Large Language Models

**Wenxuan Zhang** [1 2 *]  **Lemeng Wu** [1]  **Changsheng Zhao** [1]  **Ernie Chang** [1]  **Mingchen Zhuge** [2]  **Zechun Liu** [1]
**Andy Su** [1]  **Hanxian Huang** [1]  **Jun Chen** [1]  **Chong Zhou** [1]  **Raghuraman Krishnamoorthi** [1]  **Vikas Chandra** [1]
**Mohamed Elhoseiny** [2 †]  **Wei Wen** [1 †]

## Abstract

Diffusion Large Language Models (dLLMs) introduce a new paradigm for language generation, which in turn presents new challenges for aligning them with human preferences. In this work, we aim to improve the policy optimization for dLLMs by reducing the cost of the trajectory probability calculation, thereby enabling scaled-up offline policy training. We prove that: (i) under reference policy regularization, the probability ratio of the newly unmasked tokens is an unbiased estimate of that of intermediate diffusion states, and (ii) the probability of the full trajectory can be effectively estimated with a single forward pass of a re-masked final state. By integrating these two trajectory reduction strategies into a policy optimization objective, we propose Trajectory Reduction Policy Optimization (dTRPO). We evaluate dTRPO on 7B dLLMs across instruction-following and reasoning benchmarks. Results show that it substantially improves the core performance of state-of-the-art dLLMs, achieving gains of up to 9.6% on STEM tasks, up to 4.3% on coding tasks, and up to 3.0% on instruction-following tasks. Moreover, dTRPO exhibits strong training efficiency due to its offline, single-forward nature, and achieves improved generation efficiency through high-quality outputs. Code can be found in `https://wx-zhang.github.io/dtrpo-web/`

## 1. Introduction

Diffusion Large Language Models (dLLMs) have recently emerged as a promising language generation paradigm

---
*Work done at Meta. †Joint last author. [1]Meta AI [2]KAUST. Correspondence to: Wei Wen <wewen@meta.com>.

*Proceedings of the 43rd International Conference on Machine Learning*, Seoul, South Korea. PMLR 306, 2026. Copyright 2026 by the author(s).

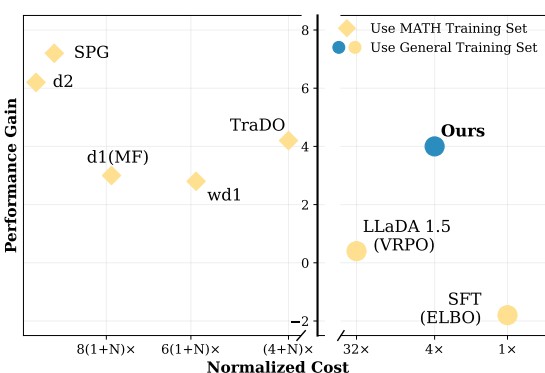

*Figure 1.* Reported performance gains on MATH dataset v.s. normalized online (left) and offline (right) training cost. dTRPO requires similar training cost to DPO for ARMs to achieve comparable performance, while other methods either incur substantially more forward passes ($32\times$) or generation steps ($N\times$).

alongside Autoregressive Large Language Models (ARMs). Inspired by discrete diffusion processes in computer vision (Song et al., 2021; Austin et al., 2021a), dLLMs formulate text generation as a discrete denoising process over a pre-defined sequence of masked tokens. This paradigm enables several distinctive capabilities, including bidirectional context awareness for tasks such as poem writing (Nie et al., 2025), controllable generation through in-place prompting (Jin et al., 2025), parallel decoding for tasks such as code generation (Xie et al., 2025), and so on.

Despite these appealing characteristics, current dLLMs still lag behind state-of-the-art ARMs (Liu et al., 2025). Years of empirical refinement have resulted in a relatively standardized post-training pipeline for ARMs (Bakouch et al., 2025; Olmo et al., 2025): supervised fine-tuning (SFT) as a cold start (Guo et al., 2025), followed by Direct Preference Optimization (DPO) (Rafailov et al., 2023) and online reinforcement learning (Bai et al., 2022) to bridge the gap between pre-training objectives and generation. Within this pipeline, the DPO stage plays an important role as an efficient and scalable intermediate step, equipping models with strong zero-shot instruction-following ability and providing a robust initialization for subsequent reinforcement learning.

Transferring this post-training paradigm to dLLMs, however, is far from straightforward. A growing body of work explores adapting post-trained ARMs to dLLMs (Ye et al., 2025; Wu et al., 2026; Gat et al., 2025; Cheng et al., 2025), as illustrated in Figure 5. Nevertheless, fundamental challenges arise in further post-training dLLMs, where the target is to improve the probability of good generation. Unlike ARMs, whose probability of generation processes can be naturally factorized at the token level, dLLMs generate text through a multi-step diffusion process over partially masked states, and estimating the probability becomes more complex. In particular, naively measuring probability of each hidden states typically requires a large number of forward passes, leading to prohibitive training costs. While recent work (Wang et al., 2025b; Zhao et al., 2026) has explored this problem in the online reinforcement learning stage, this inefficiency, together with the reliance on online training, limits the scalability of dLLMs training.

In this work, we explore how to use less computation to estimate the generation probability in the dLLM policy optimization, which we denote as Trajectory Reduction. We develop a theoretical formulation of the dLLM generation process and perform the trajectory reduction in two ways. Firstly, under reference-policy regularization, we prove that each state probability ratio admits a factorization into probability ratios over newly unmasked tokens. We then show that the trajectory probabilities can be estimated using fewer states by a single forward pass with block attention, and the estimation can be guided by the inference-time decoding strategy. Applying these reductions to the DPO objective yields Trajectory Reduction Policy Optimization (dTRPO), which enables an efficient and stable post-training procedure for diffusion language models.

We evaluate dTRPO on 7B dLLMs across instruction-following, STEM, and coding benchmarks in the zero-shot setting. Our results show that dTRPO substantially improves both instruction-following performance and reasoning capabilities, achieving gains of up to 9.6% on STEM tasks, up to 4.3% on coding tasks, and up to 3.0% on instruction-following tasks. We further analyze the effects of different implementation choices, including scheduler design, trajectory sampling strategies, and algorithm hyperparameters, and demonstrate that the proposed approach is compatible with both long-block dLLMs and block-wise dLLMs. Our contributions are

- We provide a theoretical formulation of the generation process of dLLMs and introduce dTRPO, which factorizes the required transition probability ratios into token-wise probability ratios and reduces the estimation cost to a single forward pass.

- We evaluate the proposed method on 7B dLLMs, demonstrating consistent gains in instruction-following

performance of up to 10%.

## 2. Related Works

### 2.1. Large Language Model Alignment

Bridging the gap between pre-training objectives and generation process is a critical step in the development of large language models (LLMs), and has been extensively studied in the autoregressive model (ARM) setting. Early alignment approaches relied on Supervised Fine-Tuning (SFT) and Reinforcement Learning from Human Feedback (RLHF) (Ouyang et al., 2022; Bai et al., 2022). Subsequently, Direct Preference Optimization (DPO) was introduced to convert the computationally expensive RLHF objective into a more stable and efficient supervised learning formulation (Rafailov et al., 2023). More recently, driven by the success of models such as DeepSeek-R1 and improvements in hardware efficiency, online reinforcement learning has emerged as a popular paradigm for further enhancing reasoning capabilities (Shao et al., 2024; Guo et al., 2025).

Contemporary post-training pipelines typically follow a three-stage curriculum: an initial SFT "cold start", followed by high-quality SFT or DPO as an intermediate alignment stage, and finally online reinforcement learning to further refine performance (Bakouch et al., 2025; Lambert et al., 2025). This curriculum aims to produce a strong *generic* model that generalizes across diverse downstream tasks.

### 2.2. Diffusion Language Models

Diffusion Large Language Models (dLLMs) have attracted growing attention, with research spanning pre-training (Nie et al., 2025; Ye et al., 2025; Xie et al., 2025; Song et al., 2025; Wu et al., 2026; Gat et al., 2025; Cheng et al., 2025), post-training (Zhu et al., 2025; Zhao et al., 2026; Wang et al., 2025b;a; Chen et al., 2025a; Tang et al., 2025; Wang et al., 2026), and sampling strategies (Chen et al., 2025b; Wu et al., 2025; Huang et al., 2026; Jin et al., 2025; Kim et al., 2025; Li et al., 2026). Existing training paradigms generally fall into two categories: *long-block* diffusion models trained from scratch (Arriola et al., 2025; Nie et al., 2025), and *block-wise* approaches that adapt pre-trained autoregressive checkpoints to diffusion-style generation via masked prediction (Wu et al., 2026; Gat et al., 2025; Cheng et al., 2025). Recent work converged toward the latter paradigm, as it achieves stronger performance with fewer training tokens by leveraging pre-trained knowledge.

In the post-training regime, Zhu et al. (Zhu et al., 2025) adapted DPO to dLLMs to perform supervised preference optimization. However, the majority of work has focused on policy-gradient-based optimization (Sutton et al., 1999) to improve reasoning quality on task-specific benchmarks (Zhao et al., 2026; Wang et al., 2025a;b; Tang et al.,

2025; Zhao et al., 2025; Chen et al., 2025a; Huang et al., 2026). A central technical bottleneck underlying these approaches is the estimation of probability of generation process. In ARMs, this probability naturally factorizes via causal conditioning and can be computed with a single forward pass. In contrast, dLLMs generate text through a multi-step diffusion process over partially masked states; computing exact probability of generation process typically requires expensive forward passes over intermediate states.

To address this challenge, LLaDA (Nie et al., 2025; Zhu et al., 2025) employs ELBO-based estimators with Monte Carlo sampling. d1 (Zhao et al., 2026) adopts a mean-field approximation, while TRaDO (Wang et al., 2025b) and d2 (Wang et al., 2025a) reduce variance by aggregating predictions over a subset of intermediate diffusion steps. DiffPO (Zhao et al., 2025) further introduces importance sampling by treating a reduced diffusion process as a surrogate policy. Despite these efforts, we argue that the theoretical formulation of discrete diffusion processes (Shi et al., 2024; Ou et al., 2025) is sufficient to support principled, rather than heuristic, derivations of efficient trajectory probability estimation. Building on this theoretical foundation, we present a simple and efficient approach and demonstrate its effectiveness in the following sections.

## 3. Algorithm

In this section, we first review preliminaries on dLLMs and policy optimization for ARMs in Section 3.1. We then formulate the reverse diffusion process of dLLMs as a finite-horizon Markov Decision Process (MDP) in Section 3.2, and introduce trajectory reduction in Section 3.3.

**Notation.** Let $\mathcal{V}$ be a finite vocabulary, and let $[\mathbf{M}]$ denote a special absorbing *mask* token. We define the extended alphabet as $\bar{\mathcal{V}} = \mathcal{V} \cup \{[\mathbf{M}]\}$. Let $\boldsymbol{y} \in \mathcal{V}^L$ denote an output token sequence of length $L$. The diffusion process operates on a sequence of latent states $\boldsymbol{\tau}_t \in \bar{\mathcal{V}}^L$, indexed by discrete time steps $t \in \{0, 1, \ldots, T\}$. We set $\boldsymbol{\tau}_0 = \boldsymbol{y}$ as the clean (fully unmasked) state and $\boldsymbol{\tau}_T = [[\mathbf{M}], \ldots, [\mathbf{M}]]$ as the pure-noise (fully masked) state. Throughout this section, all distributions and policies are implicitly conditioned on the prompt/context; for simplicity, we omit the prompt from the notation. Superscripts denote sequence positions, i.e., $\tau_t^{(i)}$ is the token at position $i$ at diffusion step $t$.

Many dLLMs perform diffusion block-wise, treating previously generated blocks as context. We denote $N_B$ as the number of blocks and $T_B$ as the number of steps in each block. We have $T = N_B \times T_B$. We write a state at the $s$-th block and $t$-th step as $\boldsymbol{\tau}_{s,t}$, which equals the state at global time step $sT_B + t$. In this state, all tokens before $s$-th block are unmasked and the tokens after $s$-th block are masked.

### 3.1. Preliminaries

For this subsection, we consider a single coordinate $\tau_t^{(i)} \in \bar{\mathcal{V}}$ and a single block for simplicity. Assuming independence across positions, the derivations extend to the full sequence.

**Diffusion Large Language Models.** We follow the notation of (Shi et al., 2024) to introduce the diffusion process of dLLMs, which includes forward process and reverse process. The forward process progressively corrupts the data by replacing tokens with $[\mathbf{M}]$ under the transition kernel $q$, $\boldsymbol{y} = \boldsymbol{\tau}_0 \xrightarrow{q} \ldots \xrightarrow{q} \boldsymbol{\tau}_T$ . Our goal is to learn a reverse policy that approximates the true posterior at each step: $\boldsymbol{\tau}_t \xrightarrow{\pi_\theta} \boldsymbol{\tau}_{t-1}$ . Then learned reverse policy can be used for the language generation through gradually predicting multiple tokens, as illustrated in Figure 2(a).

The transition kernel $q(\tau_t^{(i)} \mid \tau_{t-1}^{(i)})$ in the forward process is

$$q(\tau_t^{(i)} \mid \tau_{t-1}^{(i)}) = (1 - \beta_t) \mathbf{1}_{\{\tau_t^{(i)} = \tau_{t-1}^{(i)}\}} + \beta_t \mathbf{1}_{\{\tau_t^{(i)} = [\mathbf{M}]\}} , \tag{1}$$

where $\beta_t \in [0, 1]$ is the masking schedule. This implies that, in the forward process, once a token is masked, it remains masked. If a token is unmasked at time $t - 1$, then with probability $\beta_t$ it becomes masked at time $t$. The marginal distribution at time $t$ given the clean data $\boldsymbol{\tau}_0$ is

$$q(\tau_t^{(i)} \mid \tau_0^{(i)}) = \alpha_t \mathbf{1}_{\{\tau_t^{(i)} = \tau_0^{(i)}\}} + (1 - \alpha_t) \mathbf{1}_{\{\tau_t^{(i)} = [\mathbf{M}]\}} , \tag{2}$$

where $\alpha_t = \prod_{s=1}^t (1 - \beta_s)$ is the cumulative retention rate. The reverse process learns to invert this corruption. Theoretically, the exact posterior $q(\tau_{t-1}^{(i)} \mid \tau_t^{(i)}, \tau_0^{(i)})$ is :

$$q(\tau_{t-1}^{(i)} \mid \tau_t^{(i)}, \tau_0^{(i)}) = \begin{cases} 1 & \text{if } \tau_{t-1}^{(i)} = \tau_t^{(i)} = \tau_0^{(i)} , \\ \frac{\alpha_{t-1} - \alpha_t}{1 - \alpha_t} & \text{if } \tau_t^{(i)} = [\mathbf{M}], \ \tau_{t-1}^{(i)} = \tau_0^{(i)} , \\ \frac{1 - \alpha_{t-1}}{1 - \alpha_t} & \text{if } \tau_t^{(i)} = [\mathbf{M}], \ \tau_{t-1}^{(i)} = [\mathbf{M}] , \\ 0 & \text{otherwise} . \end{cases} \tag{3}$$

That is, conditioned on $\tau_t^{(i)} = [\mathbf{M}]$, with probability $\frac{1 - \alpha_{t-1}}{1 - \alpha_t}$ the token stays masked, and with probability $\frac{\alpha_{t-1} - \alpha_t}{1 - \alpha_t}$ it is unmasked; once unmasked, it remains unmasked thereafter.

To approximate the posterior, we parameterize a neural network $f_\theta(\boldsymbol{\tau}_t)$ that predicts the clean-data distribution $\boldsymbol{y} = \boldsymbol{\tau}_0$ given the current noisy state $\boldsymbol{\tau}_t$ and a conditioning context (e.g., a prompt). Let $\mu_\theta(\cdot \mid \boldsymbol{\tau}_t) = \text{softmax}(f_\theta(\boldsymbol{\tau}_t))$ denote the predicted categorical distribution over $\mathcal{V}$ at each position. The parameterized reverse transition kernel $p_\theta(\boldsymbol{\tau}_{t-1} \mid \boldsymbol{\tau}_t)$ is typically constructed by marginalizing over the predicted $\hat{\boldsymbol{\tau}}_0$ (Shi et al., 2024):

**(a) Generation Trajectory**

**(b) Trajectory Reduction**

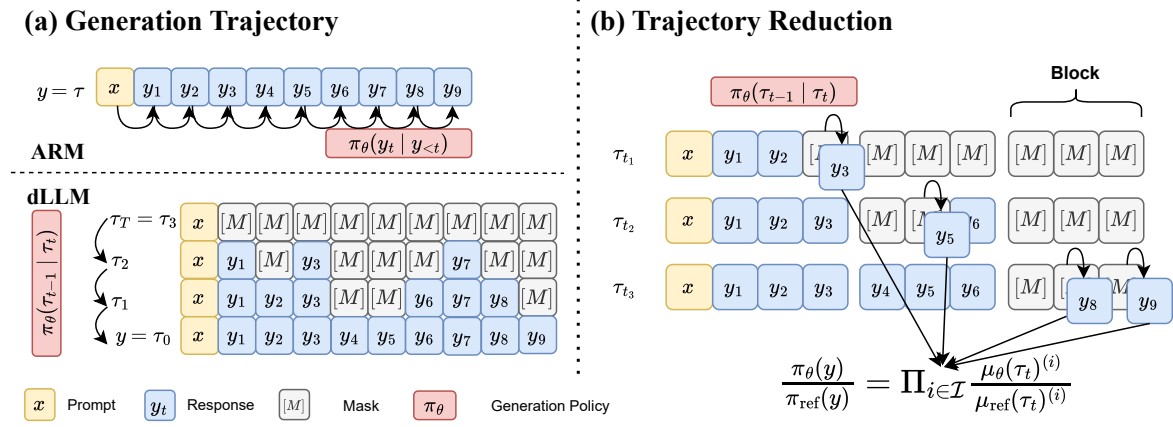

*Figure 2.* **(a):** Generation processes in ARMs and dLLMs. ARMs generate tokens via causal conditioning, whereas dLLMs generate sequences via a multi-step diffusion process. **(b):** dTRPO samples masked tokens for each block and estimates trajectory probability ratios using only the probabilities of newly unmasked tokens under $\pi_\theta$.

$$p_\theta(\tau_{t-1}^{(i)} \mid \tau_t^{(i)}, \boldsymbol{\tau}_t, \mu_\theta) =$$

$$\begin{cases} \frac{\alpha_{t-1}-\alpha_t}{1-\alpha_t}\mu_\theta(\tau_{t-1}^{(i)} \mid \boldsymbol{\tau}_t) & \tau_t^{(i)} = [\mathbf{M}], \hat{\tau}_0^{(i)} = \tau_{t-1}^{(i)} \neq [\mathbf{M}] \ , \\ \frac{1-\alpha_{t-1}}{1-\alpha_t} & \tau_t^{(i)} = [\mathbf{M}], \tau_{t-1}^{(i)} = [\mathbf{M}] \ , \\ 1 & \tau_t^{(i)} \neq [\mathbf{M}], \tau_{t-1}^{(i)} = \tau_t^{(i)} \ , \\ 0 & \text{otherwise} \ . \end{cases}$$

(4)

**Alignment with Policy Optimization in ARMs.** In the policy optimization based method, the goal of alignment is to maximize the probability of generating the completion $\boldsymbol{y} = (y_1, \ldots, y_L) \in \mathcal{V}^L$. As shown in Figure 2(a), given a prompt (context), an ARM defines a policy $\pi_\theta(y_t \mid y_{<t})$ to generate the completions and induces an MDP, whose trajectory probability factorizes as

$$\pi_\theta(\boldsymbol{y}) = \prod_{t=1}^{L} \pi_\theta(y_t \mid \boldsymbol{y}_{<t}) \ , \tag{5}$$

where $\boldsymbol{y}_{<t} = (y_1, \ldots, y_{t-1})$. Thanks to the casual attention, $\pi_\theta(y_t \mid \boldsymbol{y}_{<t})$ can be simply computed by forwarding the final sequence $\boldsymbol{y}$ and taking the probability of the token $y_t$, that is, $\pi_\theta(y_t \mid \boldsymbol{y}_{<t}) = [\pi_\theta(\boldsymbol{y})]^{(t)}$. Thus, the trajectory probability $\pi_\theta(\boldsymbol{y})$ can be computed in a single forward pass.

A class of offline alignment methods (Rafailov et al., 2023) optimizes the policy using preference data $\mathcal{D} = \{(\boldsymbol{y}^+, \boldsymbol{y}^-)\}$, where $\boldsymbol{y}^+$ is preferred over $\boldsymbol{y}^-$ under the same prompt. The objective is

$$\mathcal{L}(\theta) = -\mathbb{E}_{(\boldsymbol{y}^+, \boldsymbol{y}^-)\sim\mathcal{D}}$$
$$g\left(\lambda\left[\log\frac{\pi_\theta(\boldsymbol{y}^+)}{\pi_\theta(\boldsymbol{y}^-)} - \log\frac{\pi_{\text{ref}}(\boldsymbol{y}^+)}{\pi_{\text{ref}}(\boldsymbol{y}^-)}\right]\right) \ , \quad (6)$$

where $g$ is a projection function (e.g., log-sigmoid in DPO (Rafailov et al., 2023) and ReLU in RSO (Liu et al., 2024)), and $\lambda$ controls the KL scale.

Next, we show that dLLMs do not admit the factorization as Equation (5); consequently, the quantities needed for policy optimization must be computed over diffusion trajectories.

**3.2. Formulation of dLLMs as MDP**

Following the prior work in the field of vision diffusion models (Black et al., 2024), we cast reverse diffusion as a finite-horizon MDP. The state at reverse step $t$ is $s_t := (\boldsymbol{\tau}_t, t)$ , where $\boldsymbol{\tau}_t \in \bar{\mathcal{V}}^L$ is the partially masked sequence and $t \in \{T, T-1, \ldots, 1\}$ is the remaining diffusion time. The initial state is deterministic with $\boldsymbol{\tau}_T = [[\mathbf{M}], \ldots, [\mathbf{M}]]$. An action corresponds to selecting the next denoised sequence, $a_t := \boldsymbol{\tau}_{t-1}$ , sampled from a parameterized reverse policy $\pi_\theta$:

$$\pi_\theta(a_t \mid s_t) = \pi_\theta(\boldsymbol{\tau}_{t-1} \mid \boldsymbol{\tau}_t, t) \ . \tag{7}$$

Under the reverse process in Section 3.1, $\pi_\theta$ is instantiated by the categorical distribution induced by Equation (4). The environment transition is deterministic: $\boldsymbol{\tau}_{t-1} \leftarrow a_t$. This MDP has a fixed horizon of length $T$ and terminates at $t = 0$, yielding a completed output sequence $\boldsymbol{\tau}_0 = \boldsymbol{y}$.

Given a prompt, generation proceeds by sampling a reverse diffusion trajectory, $\boldsymbol{\tau} := (\boldsymbol{\tau}_T, \boldsymbol{\tau}_{T-1}, \ldots, \boldsymbol{\tau}_0)$ , where $\boldsymbol{\tau}_T = [[\mathbf{M}], \ldots, [\mathbf{M}]]$ and $\boldsymbol{\tau}_0 = \boldsymbol{y} \in \mathcal{V}^L$. Under $\pi_\theta$, the *trajectory probability* is

$$\pi_\theta(\boldsymbol{\tau}) = \prod_{t=1}^{T} \pi_\theta(\boldsymbol{\tau}_{t-1} \mid \boldsymbol{\tau}_t, t)$$
$$= \prod_{t=1}^{T} p_\theta(\boldsymbol{\tau}_{t-1} \mid \boldsymbol{\tau}_t, t, \mu_\theta) \ . \tag{8}$$

Since $\boldsymbol{\tau}_T$ is always fully masked, the initial-state term is constant and omitted.

## 3.3. Trajectory Reduction

In contrast to ARMs, the trajectory probability in Equation (8) does not admit a simple token-level factorization as in Equation (5). For example, in Figure 2(a), the generation of token $y_2$ depends on a partially masked state (e.g., $\boldsymbol{\tau}_2$). If we perform only a single forward pass on the fully unmasked output $\boldsymbol{y}$ (as in ARMs), the resulting conditional distribution corresponds to $\pi_\theta(\tau_1^{(2)} \mid \boldsymbol{\tau}_0)$ rather than the required transition probability $\pi_\theta(\tau_1^{(2)} \mid \boldsymbol{\tau}_2)$. As a result, trajectory probability cannot be obtained from a single forward pass. This yields the first challenge: (i) efficiently estimating trajectory probability with a small number of forward passes.

Beyond efficiency, computing the trajectory probability in Equation (8) using the kernel in Equation (4) can be numerically unstable. The schedule-dependent coefficients (involving $\alpha_t$) can differ in magnitude from the learned categorical terms $\mu_\theta(\cdot)$; with a large vocabulary, these coefficients may dominate and lead to unstable training. This yields the second challenge: (ii) handling schedule-dependent coefficients in $p_\theta$ to ensure stable training.

*Table 1.* Directly optimizing masked token probability is not effective in dLLM post training.

| Objective | GSM8K | LCBv6 | IFEval |
|---|---|---|---|
| $\max_\theta \Pi_{y_i=[\mathbf{M}]} \pi_\theta(\tilde{\boldsymbol{y}})^{(i)}$ | 66.26 | 2.37 | 25.69 |
| $\min_\theta \mathrm{ELBO}(\Pi_{y_i=[\mathbf{M}]} \pi_\theta(\tilde{\boldsymbol{y}})^{(i)})$ | 79.98 | 11.56 | 51.02 |
| dTRPO | 85.97 | 15.17 | 65.06 |

Some existing work (Zhao et al., 2026) estimates masked-token probabilities by randomly masking the final output sequence and directly optimizing the masked token probability. However, under the diffusion formulation in Section 3.2, this is not a grounded estimator of the trajectory probability in Equation (8). When used for training, other works (Ou et al., 2025) suggest it should be paired with an ELBO objective, and we show the performance in Table 1. In practice, ELBO training can be noisy and often requires multiple corrupted inputs $\tilde{\boldsymbol{y}}$ to obtain a low-variance estimate, which is still sub-optimal.

**State Reduction.** To address challenge (i), we estimate the trajectory probability in Equation (8) with a subset of timesteps. With block attention (Wu et al., 2026; Gat et al., 2025), the computation can be implemented in a single forward pass by sampling one time step $t$ within each block, details in Section B.

**Theorem 3.1** (State Reduction). *The probability of the MDP process in dLLMs can be reduced to*

$$\log \pi_\theta(\boldsymbol{\tau}) = \sum_{s=1}^{N_B} \mathbb{E}_{t\sim U(1,T_B)} T_B \log \pi_\theta(\boldsymbol{\tau}_{s,t-1} \mid \boldsymbol{\tau}_{s,t}, t) \ . \tag{9}$$

*where $\boldsymbol{\tau}_{s,t} = \boldsymbol{\tau}_{sT_B+t}$ denotes the state at block $s$ and within-block step $t$.*

The proof is provided in Section A.1. In block-wise diffusion models such as Fast-dLLM v2, $N_B$ is determined by the sequence length and block size; in long-block models such as LLaDA, one has $N_B = 1$. Globally, this estimator is unbiased and equals to using $N_B$ sampled diffusion steps.

**Ratio Reduction.** Although computing trajectory probability itself is difficult, policy optimization requires only ratios between the current policy and a reference policy. Crucially, when the two policies are evaluated on the same transition, the schedule-dependent coefficients cancel, yielding an expression depending only on the model-predicted categorical probabilities.

**Theorem 3.2** (Ratio Reduction). *Assume the reverse kernel factorizes across coordinates as in Equation (4). Then for any $(\boldsymbol{\tau}_{t-1}, \boldsymbol{\tau}_t)$,*

$$\frac{\pi_\theta(\boldsymbol{\tau}_{t-1} \mid \boldsymbol{\tau}_t, t)}{\pi_{\mathrm{ref}}(\boldsymbol{\tau}_{t-1} \mid \boldsymbol{\tau}_t, t)}$$

$$= \prod_{i\in\mathcal{I}_t(\boldsymbol{\tau}_{t-1},\boldsymbol{\tau}_t)} \frac{\mu_\theta\left(\tau_{t-1}^{(i)} \mid \boldsymbol{\tau}_t\right)}{\mu_{\mathrm{ref}}\left(\tau_{t-1}^{(i)} \mid \boldsymbol{\tau}_t\right)} \ , \tag{10}$$

*where $\mathcal{I}_t(\boldsymbol{\tau}_{t-1}, \boldsymbol{\tau}_t)$ is the set of newly unmasked coordinates at step $t$, defined by*

$$\mathcal{I}_t(\boldsymbol{\tau}_{t-1}, \boldsymbol{\tau}_t) := \left\{i \in [L] \ : \ \tau_t^{(i)} = [\mathbf{M}], \ \tau_{t-1}^{(i)} \in \mathcal{V}\right\} \ . \tag{11}$$

The proof is provided in Section A.2. We emphasize that Equation (10) holds *only* for ratios: neither $\pi_\theta(\boldsymbol{\tau}_{t-1} \mid \boldsymbol{\tau}_t, t)$ nor $\pi_{\mathrm{ref}}(\boldsymbol{\tau}_{t-1} \mid \boldsymbol{\tau}_t, t)$ alone reduces to a product of categorical terms due to the schedule coefficients and the unchanged coordinates. Therefore, it is not effective to optimize the trajectory by directly maximizing the probability of masked tokens, as the first row in Table 1.

A practical benefit of Theorem 3.2 is that the ratio in Equation (10) is independent of the masking schedule $\{\beta_t\}_{t=1}^T$: all schedule-dependent coefficients cancel between $\pi_\theta$ and $\pi_{\mathrm{ref}}$. Therefore, we may choose a scheduler that matches the base model's inference-time decoding strategy without changing the form of the loss; equivalently, we define $\mathcal{I}_t$ according to the inference-time unmasking strategy.

Here, the *scheduler* specifies (i) the number of reverse steps $T$ and (ii) how tokens are unmasked at each step. We adopt an inference-aligned scheduler: at each reverse step, we unmask $k = 0.1 \times$ (block size) tokens by selecting the top-$k$ masked coordinates according to the confidence scores. For example, in Figure 2(b), if $\mathcal{I}_t = \{2, 4\}$, we use only these two categorical probabilities to form the trajectory ratio at

that step. Although selecting top-$k$ coordinates violates strict conditional independence, we empirically show in Section 4.4 that the results are negligible impacted.

**dTRPO Objective.** Apply Theorem 3.1 and Theorem 3.2 in DPO (Rafailov et al., 2023), dTRPO is formulated as:

$$\mathcal{L}_{\text{dTRPO}}(\theta) = -\mathbb{E}_{(\boldsymbol{y}^+, \boldsymbol{y}^-) \sim \mathcal{D}} \log \sigma$$
$$\left( \lambda T_B \left[ S(\boldsymbol{y}_{s,t}^+; \theta, \text{ref}) - S(\boldsymbol{y}_{s,t}^-; \theta, \text{ref}) \right] \right) , \quad (12)$$

where the term $S(\boldsymbol{y}; \theta, \text{ref})$ is the log of probability ratios over the newly unmasked tokens for summed over blocks

$$S(\boldsymbol{y}; \theta, \text{ref}) = \sum_{s=1}^{N_B} \mathbb{E}_{t \sim U(1, T_B)}$$
$$\log \left[ \prod_{i \in \mathcal{I}_t(\boldsymbol{\tau}_{s,t-1}, \boldsymbol{\tau}_{s,t})} \frac{\mu_\theta \left( \tau_{s,t-1}^{(i)} \mid \boldsymbol{\tau}_{s,t} \right)}{\mu_{\text{ref}} \left( \tau_{s,t-1}^{(i)} \mid \boldsymbol{\tau}_{s,t} \right)} \right] . \quad (13)$$

and $\mathcal{I}_t(\boldsymbol{\tau}_{s,t-1}, \boldsymbol{\tau}_{s,t})$ is the set of newly unmasked coordinates at block $s$ and within-block step $t$, selected by top-$k$ confidence during the training. The derivation is provided in Section A.4. We further provide a bias and variance analysis in Section A.3. Algorithms can be found in Algorithm 1. Moreover, our theorems also generalize to policy-gradient methods, as detailed in Section A.5.

### 3.4. Relation to Early Works

Early work (Zhao et al., 2026; Tang et al., 2025; Chen et al., 2025a; Wang et al., 2025a; Zhao et al., 2025) commonly optimizes the KL-regularized objective using policy-gradient updates under a mean-field approximation. These methods typically construct a stochastic estimator by (i) sampling and forwarding several reverse steps and randomly masking tokens, and (ii) approximating the objective using the model predictions on the masked positions. In contrast, we estimate the required ratios using $N_B$ sampled steps (one per block) with one forward pass. Moreover, we do not use the probabilities of all masked tokens; instead, we use only the probabilities of newly unmasked tokens.

The above procedure can be interpreted as a special case of our framework. For long-block models (where $N_B = 1$), it corresponds to a scheduler that, at the sampled step, unmasks *all* currently masked coordinates (i.e., $k$ equals the number of masked tokens at that step). Since these works employ policy-gradient objectives that depend on policy ratios, our ratio-reduction theorem also provides a principled explanation of their estimators.

In practice, such a scheduler is often suboptimal, both in optimization stability and in matching the model's deployment-time generation dynamics, which we verify empirically in our experiments.

## 4. Experiments

### 4.1. Setup

**Backbone Model and Inference Strategy.** We select Fast-dLLM-v2-7B as the backbone model due to its strong inference efficiency and native support for parallel decoding (Wu et al., 2026). Fast-dLLM-v2-7B is adapted from Qwen2.5-7B-Instruct (Qwen et al., 2025) and employs a block-wise diffusion process. At inference time, generation proceeds block by block (with size 32) in an autoregressive manner. We follow the official Fast-dLLM-v2 inference implementation, using greedy decoding with a batch size of 32 and a maximum generation length of 2048 tokens.

**Training Details.** Following SmolLM3, we primarily use the SmolTalk2 preference dataset to improve instruction-following capability (Bakouch et al., 2025). In addition, we mix it with a math preference dataset (Argilla, 2024) and a code preference dataset (Vezora, 2024) to enhance mathematical and code reasoning performance. The total number of preference pairs is 500K.

We train the model for one epoch with a per-device batch size of 2 and a gradient accumulation factor of 8 across 64 NVIDIA A100 (80GB) GPUs; total training takes approximately 5 hours. Optimization is performed using AdamW (Loshchilov & Hutter, 2019) with a learning rate of $5 \times 10^{-7}$. We employ a cosine-annealing learning rate schedule with a warmup phase covering the first 10% of training steps. The context length during training is set to 4096 tokens. For the remaining hyperparameters, we follow the training script of SmolLM3 (Bakouch et al., 2025).

For dTRPO, we use a top-$k$ confidence scheduler with $k = 0.1$. The training block size is 32, which corresponds to selecting 3 newly unmasked tokens per block to form probability ratios. For preference optimization, we use the DPO (Rafailov et al., 2023) objective with $\lambda = 0.05$. To reduce training cost, we follow BFPO (Zhang et al., 2025) and update only the MLP layers and the output projection layers, while keeping the remaining parameters frozen. More training details can be found in Section D.1.

**Evaluation Benchmarks.** We evaluate models on instruction-following, math and STEM reasoning, and code generation benchmarks, following the evaluation protocols of SmolLM3 (Bakouch et al., 2025) and Fast-dLLM-v2 (Wu et al., 2026). For math reasoning, we report results on GPQA (Rein et al., 2024), GSM8K (Cobbe et al., 2021), and MATH (Hendrycks et al., 2021). For code generation, we evaluate on MBPP (Austin et al., 2021b), LiveCodeBench v6 (LCBv6) (Jain et al., 2025), and HumanEval (Chen, 2021). For instruction following, we use IFEval (Zhou et al., 2023), Arena-Hard v2 (Li et al., 2025), and MT-Bench (Zheng et al., 2023). All benchmarks are evaluated in the *zero-shot* setting. Detailed descriptions of datasets

*Table 2.* Performance of dLLMs under **zero-shot** evaluation. dTRPO achieves overall best performance and close the gap to strong ARMs.

| | GPQA (cot,diamond) | GSM8K (cot) | MATH | LCB (v6) | MBPP+ (extra) | HEval+ (extra) | IFEval (prompt) | ArenaHard (V2.0) | MTBench |
|---|---|---|---|---|---|---|---|---|---|
| *dLLM from Scratch* | | | | | | | | | |
| LLaDA Instruct | 19.19 | 78.47 | 42.48 | 6.07 | 38.1 | 34.1 | 53.23 | - | - |
| LLaDA 1.5 | 19.19 | 79.45 | 43.64 | 6.54 | 37 | 39 | 59.52 | - | - |
| *Qwen 2.5 7B Instruct v.s. dLLM from Qwen2.5 7B Instruct* | | | | | | | | | |
| Qwen2.5 Instruct | 36.36 | 87.87 | 73.06 | 24.42 | 67.5 | 74.4 | 71.38 | 10.43 | 8.08 |
| Dream Instruct | 28.79 | 75.36 | 50.22 | 12.61 | **54.5** | 53 | 50.65 | 6.79 | 3.88 |
| Fast-dLLM-v2 | 20.71 | 82.34 | 60.26 | 11.56 | 51.6 | 59.1 | 62.11 | 1.26 | 3.17 |
| Fast-dLLM-v2+ELBO | 12.63 | 79.98 | 58.48 | 11.56 | 52.4 | 59.1 | 51.02 | 0.17 | 1.01 |
| Fast-dLLM-v2+VRPO | 24.24 | 83.17 | 63.32 | 12.89 | 50.5 | 57.3 | 65.06 | 7.32 | 6.37 |
| Fast-dLLM-v2+DPO w/ MF | 23.74 | 85.37 | 63.20 | 11.00 | 46.30 | 51.80 | **65.62** | 6.02 | 6.48 |
| **Fast-dLLM-v2+dTRPO** | **30.30** | **85.97** | **64.3** | **15.17** | 51.6 | **63.4** | 65.06 | **7.41** | **6.53** |

and evaluation metrics are provided in Section D.3.

**Comparison Baselines.** We compare the following 7B diffusion language models initialized from Qwen2.5-7B-Instruct using their official checkpoints: LLaDA (Nie et al., 2025), Dream (Ye et al., 2025), and Fast-dLLM-v2 (Wu et al., 2026). We additionally compare dTRPO against several alternative estimators, including ELBO-based supervised fine-tuning (Nie et al., 2025), VRPO (Zhu et al., 2025), and DPO with a block-wise mean-field approximation (DPO w/ MF) (Zhao et al., 2026). All alternative estimators are trained with the same data and training configuration as dTRPO. Detailed information are in Section D.4.

### 4.2. Results Overview

Evaluation results are summarized in Table 2. Across all evaluated benchmarks, dTRPO delivers the most robust overall performance among the open-source dLLM baselines. Specifically, compared to the strongest dLLM baseline, Fast-dLLM-v2, dTRPO achieves significant performance gains: 9.59% on GPQA, 3.63% on GSM8K, 4.04% on MATH, and 3.61% on LCBv6. Furthermore, it enhances code generation capabilities (HumanEval+ by 4.3%) and instruction-following proficiency (IFEval by 2.95%). For open-ended benchmarks evaluated under the LLM-as-a-judge protocol, dTRPO outperforms Fast-dLLM-v2 by 6.15% on Arena-Hard and 3.36% on MT-Bench.

Notably, we implemented several representative post-training strategies for comparison, and dTRPO consistently outperforms these alternatives in aggregate. While certain baselines achieve localized advantages on specific benchmarks—for instance, Dream leads on MBPP+ by 2.9% and DPO w/ MF slightly edges out on IFEval by 0.56%—these instances likely stem from a specialized focus within their respective training data distributions. However, these methods typically fall substantially behind on other critical benchmarks; for example, Dream lags behind dTRPO by 14.08% on MATH. In contrast, dTRPO maintains a balanced and

*Table 3.* Tokens per second (TPS) and task performance comparison on GSM8K and Arena-Hard between dTRPO, Fast-dLLM-v2, and Qwen2.5-7B-Instruct.

| Model | GSM8K | | Arena-Hard | |
|---|---|---|---|---|
| | TPS | Accuracy | TPS | Score |
| Qwen2.5-7B-Instruct | 38.9 | 87.87 | 16.20 | 10.43 |
| Fast-dLLM-v2 | 38.84 | 82.34 | 19.55 | 6.79 |
| dTRPO | 38.80 | 85.97 | 29.87 | 7.41 |

superior performance profile across the entire suite.

Moreover, dTRPO effectively narrows the performance gap between dLLMs and strong ARMs, such as Qwen2.5-7B-Instruct. Since most dLLMs are adapted from ARMs, the ability to preserve the original pre-trained knowledge is paramount for maintaining complex reasoning tasks. While some performance degradation relative to the base ARM is an inherent trade-off for the significantly higher sampling efficiency of dLLMs ($1.9 \times$ in Figure 6(d)), the drop exhibited by dTRPO is relatively marginal. This is especially evident in instruction-following tasks evaluated by LLM-as-a-judge, such as Arena-Hard and MT-Bench, where the performance gaps are narrowed to a nearly negligible 3.02% and 1.55%, respectively. We provide more comparisons with other model backbones in Section E.1.

### 4.3. Training and Inference Efficiency

As shown in Figure 1, our training cost matches that of DPO for ARMs while achieving comparable MATH performance to dLLM baselines that are trained with MATH-specific data. Concretely, for each training example, we forward the preferred and dispreferred completions once under the policy model and once under the reference model, for a total of 4 forward passes, which follows the DPO training paradigm for ARMs. For online methods such as TRaDO or d1, training additionally requires generating completions, which equals hundreds of forward passes (cost $N \times$), and for each completion, multiple forward passes are typically

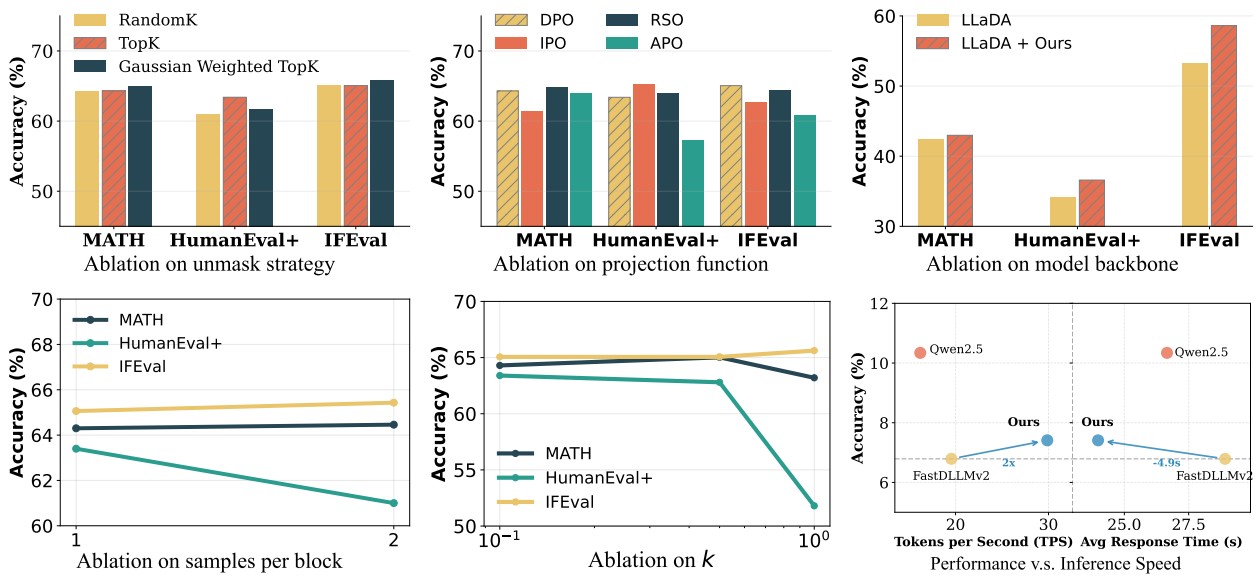

*Figure 3.* Ablation study of algorithm design and implementation choices; inference speed comparison for dTRPO and baseline models.

needed to estimate trajectory probability ratios. Overall, dTRPO is purely offline and requires training compute comparable to DPO for ARMs.

To evaluate inference efficiency, we report tokens per second (TPS) and average end-to-end inference time on GSM8K and Arena-Hard. We compare dTRPO, Fast-dLLM-v2, and Qwen2.5-7B-Instruct in Table 3 and Figure 3. As shown in Table 3, on easier benchmarks like GSM8K, dLLMs generally achieve higher TPS than ARMs. dTRPO attains efficiency comparable to Fast-dLLM-v2 while achieving substantial improvements in accuracy. On Arena-Hard, dLLMs still lag behind ARMs in overall performance. The weaker capabilities of dLLMs can lead to redundant generation, which may increase inference time. Nevertheless, dTRPO achieves the highest TPS and the lowest inference time, while narrowing the performance gap between dLLMs and ARMs.

### 4.4. Ablations and Discussion

**Scheduling Strategy.** Our method adopts a confidence-based scheduling strategy that unmasks tokens with the highest confidence at each diffusion step. However, the derivation in Theorem 3.1 relies on an independence assumption across token positions. To examine sensitivity to this assumption, we also evaluate a *random-k* scheduler, which unmasks a randomly selected $k\%$ of masked tokens at each step, independent of confidence scores. We further design a *Gaussian-weighted top-k* scheduler, which reweights confidence scores by a Gaussian function over token indices, assigning higher weights to earlier tokens in the sequence. We use $k = 0.1$ for all schedulers.

As shown in Figure 3, the differences among scheduling strategies are generally small, and each strategy exhibits varying strengths across tasks. In particular, for instruction-following benchmarks, the Gaussian-weighted top-$k$ scheduler outperforms the alternatives. This suggests that prioritizing earlier tokens when estimating *sequence probability ratios* can be beneficial in instruction-following settings.

**Number of Time Steps per Block.** In our experiments, we set $k = 0.1$, which corresponds to $T_B = 10$ time steps per block. We ablate the number of time steps per block by varying $k$. As shown in Figure 3(f), larger $k$ implies fewer time steps per block and leads to worse performance. In practice, inference-time decoding typically unmasks only a small number of tokens per step (often starting from one token and increasing toward the end), which supports using small $k$. Overall, these results indicate that our estimator remains effective under inference-aligned unmasking schedules.

**Number of Samples per Block.** Our method estimates the required trajectory probability ratios using a single sampled diffusion step per block. We ablate the number of sampled steps in Figure 3(e). As shown, increasing the number of sampled steps yields comparable performance, indicating that the estimator remains stable even with a single sample.

**dTRPO on LLaDA.** We further apply dTRPO to LLaDA, which employs a single long diffusion block where each token attends to all other tokens. As shown in Figure 3(c), dTRPO yields a substantial improvement over the LLaDA backbone. This indicates that the proposed approach is not limited to block-wise diffusion architectures and also extends effectively to long-block dLLMs.

**Projection Function.** We use the DPO (Rafailov et al.,

2023) log-sigmoid projection function to map probability ratios into the loss. Given extensive work on projection functions for preference optimization, we evaluate several commonly used alternatives, including DPO, IPO (Azar et al., 2024), RSO (Liu et al., 2024), and APO (Doosterlinck et al., 2025). As shown in Figure 3(b), our estimator is robust to the choice of projection function. While different projections exhibit task-dependent differences, the log-sigmoid projection performs best on instruction-following benchmarks. See Section E.2 for the ablations of DPO $\lambda$ and parameter efficient training

## 5. Conclusion

We studied a problem in alignment for diffusion large language models (dLLMs): computing the trajectory probability is substantially more expensive and less stable than the token-factorized sequence probability used by autoregressive models. To address this, we introduced dTRPO, which combines *state reduction* and *ratio reduction* to obtain an efficient estimator that uses only the probabilities of newly unmasked tokens and can be implemented with a single forward pass per block. Experiments on 7B dLLMs show that dTRPO consistently improve instruction-following, STEM reasoning, and coding performance, outperforming representative alternative estimators and other open-source dLLMs. We hope these findings motivate future work to scale up the post-training stage of dLLMs.

## Impact Statement

This paper presents work whose goal is to advance the field of Machine Learning. There are many potential societal consequences of our work, none which we feel must be specifically highlighted here.

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

## Table of Contents

## A. Theorem

### A.1. Proof of Theorem 3.1 (State Reduction)

*Proof.* By the definition of the MDP trajectory probability in Equation (8), the log-probability of a reverse diffusion trajectory $\boldsymbol{\tau}$ is a sum over transition log-probabilities:

$$\log \pi_\theta(\boldsymbol{\tau}) = \sum_{t=1}^{T} \log \pi_\theta(\boldsymbol{\tau}_{t-1} \mid \boldsymbol{\tau}_t, t). \tag{14}$$

We partition the horizon into $N_B$ blocks of equal length $T_B$ such that $T = N_B T_B$. Let $\boldsymbol{\tau}_{s,t}$ denote the state at block $s \in \{1, \ldots, N_B\}$ and within-block step $t \in \{1, \ldots, T_B\}$ (i.e., global time index $sT_B + t$). Then we can re-index the sum as

$$\log \pi_\theta(\boldsymbol{\tau}) = \sum_{s=1}^{N_B} \sum_{t=1}^{T_B} \log \pi_\theta(\boldsymbol{\tau}_{s,t-1} \mid \boldsymbol{\tau}_{s,t}, t). \tag{15}$$

For any function $f(t)$ over $\{1, \ldots, T_B\}$, we have the identity $\sum_{t=1}^{T_B} f(t) = T_B \, \mathbb{E}_{t \sim U(1,T_B)}[f(t)]$. Applying this to the inner sum yields

$$\log \pi_\theta(\boldsymbol{\tau}) = \sum_{s=1}^{N_B} T_B \, \mathbb{E}_{t \sim U(1,T_B)} \Big[\log \pi_\theta(\boldsymbol{\tau}_{s,t-1} \mid \boldsymbol{\tau}_{s,t}, t)\Big]. \tag{16}$$

Therefore, sampling a single time step within each block provides an unbiased estimator of the full-horizon log-probability, while reducing the number of required evaluations from $T$ to $N_B$. $\qquad\square$

## A.2. Proof of Theorem 3.2 (Ratio Reduction)

*Proof.* Assume the reverse kernel factorizes across coordinates as in Equation (4). Then

$$\pi_\theta(\boldsymbol{\tau}_{t-1} \mid \boldsymbol{\tau}_t, t) = \prod_{i=1}^{L} p_\theta(\tau_{t-1}^{(i)} \mid \tau_t^{(i)}, \boldsymbol{\tau}_t, \mu_\theta). \tag{17}$$

Fix a transition $(\boldsymbol{\tau}_{t-1}, \boldsymbol{\tau}_t)$. Partition indices $i \in [L]$ into three disjoint sets:

1. **Unmasked/Context positions:** $\mathcal{U} := \{i : \tau_t^{(i)} \in \mathcal{V}\}$. By Equation (4), such positions are deterministic and contribute a factor 1.

2. **Newly unmasked positions:** $\mathcal{I}_t := \{i : \tau_t^{(i)} = [\mathbf{M}], \ \tau_{t-1}^{(i)} \in \mathcal{V}\}$. By Equation (4),

$$p_\theta = \frac{\alpha_{t-1} - \alpha_t}{1 - \alpha_t} \, \mu_\theta(\tau_{t-1}^{(i)} \mid \boldsymbol{\tau}_t).$$

3. **Still masked positions:** $\mathcal{M} := \{i : \tau_t^{(i)} = [\mathbf{M}], \ \tau_{t-1}^{(i)} = [\mathbf{M}]\}$. By Equation (4),

$$p_\theta = \frac{1 - \alpha_{t-1}}{1 - \alpha_t},$$

which is schedule-dependent but model-independent.

Now consider the ratio between the current policy and the reference policy, evaluated on the *same* transition $(\boldsymbol{\tau}_{t-1}, \boldsymbol{\tau}_t)$:

$$\frac{\pi_\theta(\boldsymbol{\tau}_{t-1} \mid \boldsymbol{\tau}_t, t)}{\pi_{\mathrm{ref}}(\boldsymbol{\tau}_{t-1} \mid \boldsymbol{\tau}_t, t)} = \prod_{i \in \mathcal{U}} \frac{1}{1} \cdot \prod_{i \in \mathcal{I}_t} \frac{\frac{\alpha_{t-1} - \alpha_t}{1 - \alpha_t} \mu_\theta(\tau_{t-1}^{(i)} \mid \boldsymbol{\tau}_t)}{\frac{\alpha_{t-1} - \alpha_t}{1 - \alpha_t} \mu_{\mathrm{ref}}(\tau_{t-1}^{(i)} \mid \boldsymbol{\tau}_t)} \cdot \prod_{i \in \mathcal{M}} \frac{\frac{1 - \alpha_{t-1}}{1 - \alpha_t}}{\frac{1 - \alpha_{t-1}}{1 - \alpha_t}}. \tag{18}$$

All schedule-dependent coefficients cancel, leaving

$$\frac{\pi_\theta(\boldsymbol{\tau}_{t-1} \mid \boldsymbol{\tau}_t, t)}{\pi_{\mathrm{ref}}(\boldsymbol{\tau}_{t-1} \mid \boldsymbol{\tau}_t, t)} = \prod_{i \in \mathcal{I}_t} \frac{\mu_\theta(\tau_{t-1}^{(i)} \mid \boldsymbol{\tau}_t)}{\mu_{\mathrm{ref}}(\tau_{t-1}^{(i)} \mid \boldsymbol{\tau}_t)}. \tag{19}$$

Thus, the transition-level policy ratio depends only on categorical probabilities at newly unmasked positions and is independent of the masking schedule. $\square$

## A.3. Bias and Variance Analysis of State Reduction

We analyze the bias and variance introduced by the block-wise estimator when approximating the sum of transition log-ratios along a fixed trajectory.

Let $\Delta_{s,t}$ denote the transition log-ratio at block $s$ and within-block step $t$:

$$\Delta_{s,t} := \log \frac{\pi_\theta(\boldsymbol{\tau}_{s,t-1} \mid \boldsymbol{\tau}_{s,t}, t)}{\pi_{\mathrm{ref}}(\boldsymbol{\tau}_{s,t-1} \mid \boldsymbol{\tau}_{s,t}, t)}. \tag{20}$$

By Theorem 3.2, this quantity is schedule-independent and can be written as

$$\Delta_{s,t} = \sum_{i \in \mathcal{I}_t(\boldsymbol{\tau}_{s,t-1}, \boldsymbol{\tau}_{s,t})} \log \frac{\mu_\theta(\tau_{s,t-1}^{(i)} \mid \boldsymbol{\tau}_{s,t})}{\mu_{\mathrm{ref}}(\tau_{s,t-1}^{(i)} \mid \boldsymbol{\tau}_{s,t})}. \tag{21}$$

Define the full trajectory objective (sum over all blocks and steps) as

$$L := \sum_{s=1}^{N_B} \sum_{t=1}^{T_B} \Delta_{s,t}. \tag{22}$$

Our estimator samples one step $t_s \sim U(1, T_B)$ independently for each block $s$ and uses

$$\hat{L} := \sum_{s=1}^{N_B} \hat{\ell}_s, \qquad \hat{\ell}_s := T_B \cdot \Delta_{s, t_s}. \tag{23}$$

**Proposition A.1** (Unbiasedness). *$\hat{L}$ is an unbiased estimator of $L$, i.e., $\mathbb{E}[\hat{L}] = L$.*

*Proof.* By linearity of expectation,

$$\mathbb{E}[\hat{L}] = \sum_{s=1}^{N_B} \mathbb{E}_{t_s \sim U(1, T_B)}[T_B \Delta_{s, t_s}] = \sum_{s=1}^{N_B} \sum_{t=1}^{T_B} \Delta_{s, t} = L.$$

$\square$

**Proposition A.2** (Variance). *Let $\sigma_s^2 := \mathrm{Var}_{t \sim U(1, T_B)}(\Delta_{s, t})$. Then*

$$\mathrm{Var}(\hat{L}) = \sum_{s=1}^{N_B} T_B^2 \sigma_s^2. \tag{24}$$

*Proof.* Independence across blocks implies $\mathrm{Var}(\hat{L}) = \sum_{s=1}^{N_B} \mathrm{Var}(\hat{\ell}_s)$. For each block, $\mathrm{Var}(\hat{\ell}_s) = \mathrm{Var}(T_B \Delta_{s, t_s}) = T_B^2 \mathrm{Var}(\Delta_{s, t_s}) = T_B^2 \sigma_s^2$. Summing over $s$ yields the claim. $\square$

This result highlights a compute–variance trade-off: the estimator remains unbiased while its variance scales as $T_B^2$. In practice, block-wise dLLMs typically use small block sizes (e.g., up to 32 tokens) (Wu et al., 2026; Cheng et al., 2025), which keeps the variance moderate while enabling substantial compute savings.

### A.4. Derivation of the dTRPO Objective

To derive the dTRPO objective, we start with the Bradley-Terry preference model and the standard DPO framework (Rafailov et al., 2023). For a diffusion trajectory $\boldsymbol{\tau}$, the relationship between the optimal reverse policy $\pi_\theta$ and the underlying reward $r(\boldsymbol{y})$ is given by $\log \frac{\pi_\theta(\boldsymbol{\tau})}{\pi_{\mathrm{ref}}(\boldsymbol{\tau})} = \frac{1}{\lambda} r(\boldsymbol{y}) - \log Z$, where $Z$ is the partition function, only depends on the context. Substituting this into the preference probability $P(\boldsymbol{y}^+ \succ \boldsymbol{y}^-) = \sigma(r(\boldsymbol{y}^+) - r(\boldsymbol{y}^-))$ yields the trajectory-based DPO objective:

$$-\mathbb{E}_{(\boldsymbol{y}^+, \boldsymbol{y}^-) \sim \mathcal{D}} \log \sigma \left( \lambda \log \frac{\pi_\theta(\boldsymbol{\tau}^+)}{\pi_{\mathrm{ref}}(\boldsymbol{\tau}^+)} - \lambda \log \frac{\pi_\theta(\boldsymbol{\tau}^-)}{\pi_{\mathrm{ref}}(\boldsymbol{\tau}^-)} \right). \tag{25}$$

Using Theorem 3.1, we decompose the global trajectory log-ratio into a sum of block-wise expectations over the $N_B$ blocks:

$$\log \frac{\pi_\theta(\boldsymbol{\tau})}{\pi_{\mathrm{ref}}(\boldsymbol{\tau})} = \sum_{s=1}^{N_B} \mathbb{E}_{t \sim U(1, T_B)} T_B \left[ \log \pi_\theta(\boldsymbol{\tau}_{s, t-1} \mid \boldsymbol{\tau}_{s, t}, t) - \log \pi_{\mathrm{ref}}(\boldsymbol{\tau}_{s, t-1} \mid \boldsymbol{\tau}_{s, t}, t) \right]. \tag{26}$$

Here we sample the same time step $t$ within each block for both the policy and reference policy. By substituting this estimation into the trajectory-based objective, the loss becomes:

$$-\mathbb{E}_{\mathcal{D}} \log \sigma \left( \sum_{s=1}^{N_B} \lambda T_B \mathbb{E}_{t \sim U(1, T_B)} \log \frac{\pi_\theta(\boldsymbol{\tau}_{s, t-1}^+ \mid \boldsymbol{\tau}_{s, t}^+, t)}{\pi_{\mathrm{ref}}(\boldsymbol{\tau}_{s, t-1}^+ \mid \boldsymbol{\tau}_{s, t}^+, t)} - \sum_{s=1}^{N_B} \lambda T_B \mathbb{E}_{t \sim U(1, T_B)} \log \frac{\pi_\theta(\boldsymbol{\tau}_{s, t-1}^- \mid \boldsymbol{\tau}_{s, t}^-, t)}{\pi_{\mathrm{ref}}(\boldsymbol{\tau}_{s, t-1}^- \mid \boldsymbol{\tau}_{s, t}^-, t)} \right). \tag{27}$$

On this basis, we leverage the **Ratio Reduction** property derived in Theorem 3.2. As shown in (10), for any single transition $(\boldsymbol{\tau}_{s, t-1}, \boldsymbol{\tau}_{s, t})$, all schedule-dependent coefficients cancel out in the ratio, simplifying the transition-level log-ratio to:

$$\log \frac{\pi_\theta(\boldsymbol{\tau}_{s, t-1} \mid \boldsymbol{\tau}_{s, t}, t)}{\pi_{\mathrm{ref}}(\boldsymbol{\tau}_{s, t-1} \mid \boldsymbol{\tau}_{s, t}, t)} = \sum_{i \in \mathcal{I}_t(\boldsymbol{\tau}_{s, t-1}, \boldsymbol{\tau}_{s, t})} \log \frac{\mu_\theta(\tau_{s, t-1}^{(i)} \mid \boldsymbol{\tau}_{s, t})}{\mu_{\mathrm{ref}}(\tau_{s, t-1}^{(i)} \mid \boldsymbol{\tau}_{s, t})}. \tag{28}$$

Finally, by substituting this simplified ratio back into (26), we arrive at the dTRPO objective presented in Section 3.3:

$$\mathcal{L}_{\text{dTRPO}}(\theta) = -\mathbb{E}_{\mathcal{D}} \log \sigma \left( \sum_{s=1}^{N_B} \mathbb{E}_{t \sim U(1,T_B)} \lambda T_B \sum_{i \in \mathcal{I}_t^+} \log \frac{\mu_\theta(\tau_{s,t-1}^{+,(i)} \mid \boldsymbol{\tau}_{s,t}^+)}{\mu_{\text{ref}}(\tau_{s,t-1}^{+,(i)} \mid \boldsymbol{\tau}_{s,t}^+)} - \sum_{s=1}^{N_B} \mathbb{E}_{t \sim U(1,T_B)} \lambda T_B \sum_{i \in \mathcal{I}_t^-} \log \frac{\mu_\theta(\tau_{s,t-1}^{-,(i)} \mid \boldsymbol{\tau}_{s,t}^-)}{\mu_{\text{ref}}(\tau_{s,t-1}^{-,(i)} \mid \boldsymbol{\tau}_{s,t}^-)} \right).$$
$$(29)$$

where $\mathcal{I}_t$ denotes the set of newly unmasked coordinates in each trajectory, and $t$ is a randomly sampled time step within each block.

### A.5. Generalization to Policy Gradient Methods

The reduction properties in Theorems 3.1 and 3.2 also apply to policy-gradient-style objectives that depend on likelihood ratios between the current policy and a reference (or old) policy.

Consider a KL-regularized RL objective of the form

$$J(\theta) = \mathbb{E}_{\boldsymbol{\tau} \sim \pi_\theta} \left[ R(\boldsymbol{\tau}) - \lambda \, \text{KL}\big(\pi_\theta(\boldsymbol{\tau}) \,\|\, \pi_{\text{ref}}(\boldsymbol{\tau})\big) \right], \tag{30}$$

where $R(\boldsymbol{\tau})$ is a trajectory-level reward and $\pi_{\text{ref}}$ is a fixed reference policy. A standard score-function (REINFORCE) gradient estimator is

$$\nabla_\theta J(\theta) = \mathbb{E}_{\boldsymbol{\tau} \sim \pi_\theta} \left[ \nabla_\theta \log \pi_\theta(\boldsymbol{\tau}) \cdot A(\boldsymbol{\tau}) \right], \tag{31}$$

where $A(\boldsymbol{\tau})$ is an advantage estimate.

In PPO-style surrogates, one typically uses a likelihood ratio $r(\boldsymbol{\tau}) := \frac{\pi_\theta(\boldsymbol{\tau})}{[\pi_{\text{ref}}(\boldsymbol{\tau})]_{\text{sg}}}$ (with stop-gradient on the denominator) and optimizes objectives of the form

$$\mathcal{L}_{\text{PG}}(\theta) = \mathbb{E}_{\boldsymbol{\tau} \sim \pi_\theta} \left[ r(\boldsymbol{\tau}) \cdot A(\boldsymbol{\tau}) \right]. \tag{32}$$

For dLLMs, $\pi_\theta(\boldsymbol{\tau})$ decomposes over reverse diffusion transitions as in Equation (8), and direct evaluation of $r(\boldsymbol{\tau})$ is expensive. By Theorem 3.1, the trajectory-level log-ratio can be estimated block-wise by sampling one step per block, and by Theorem 3.2, each sampled transition ratio cancels all schedule-dependent coefficients. Therefore, the same reductions can be used to compute PPO/DPO-style ratios with substantially fewer forward passes.

## B. Block Attention

This section explains how the expectation in Equation (9) can be computed with a single forward pass. The key is a customized training-time attention mask that allows different tokens within the same packed sequence to observe different contexts, effectively simulating multiple partially masked states (one per block) in parallel.

**Causal attention in ARMs.** As shown in Figure 4(a), autoregressive (ARM) training uses a *causal* mask so that token $i$ attends only to tokens $< i$ (and the prompt). Consequently, a single forward pass on the full sequence yields token-wise probabilities $\pi_\theta(y_i \mid y_{<i})$, matching the factorization in Equation (5).

**Why this fails for dLLMs.** For dLLMs, the relevant quantity in Equation (8) is the transition probability conditioned on a *partially masked* state. At training time, for token positions in block $s$, we need probabilities conditioned on: (i) all previous blocks being clean, and (ii) the current block being partially masked. A single standard forward pass on one corrupted sequence cannot simultaneously provide the correct conditioning for all blocks.

**Block attention mask.** In Figure 4(b), we show a packed sequence that concatenates partially masked tokens with the corresponding clean tokens, and define a block attention mask such that: (i) each token in a block attends to clean tokens from previous blocks, and (ii) masked tokens in the current block attend to the partially masked tokens within the same block. This design ensures that the logits for masked positions are conditioned on the desired "middle state" for each block.

With this mask, a single forward pass is equivalent to forwarding $N_B$ separate partially masked sequences (one per block state). By extracting logits from the masked half, we obtain exactly the per-block sampled transition terms required by Equation (9). If multiple time steps per block are needed, we can resample masks and repeat the forward pass accordingly.

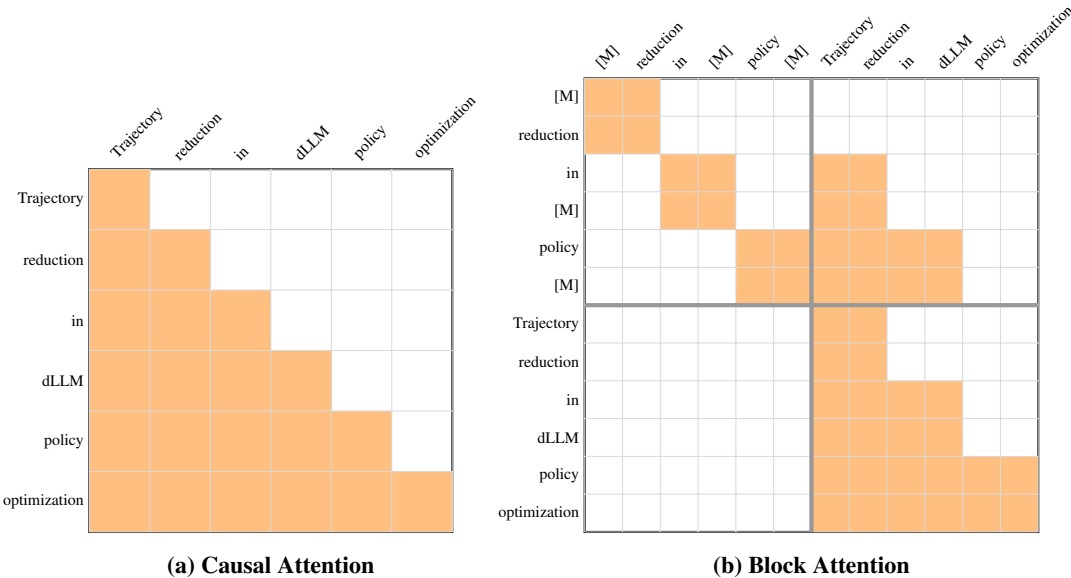

(a) **Causal Attention**  (b) **Block Attention**

*Figure 4.* Causal attention and block attention in the training time. In (b), each row represents the tokens that an input token attends to. For a masked token, it attends to the masked input of the same block and clean tokens in the previous blocks.

## C. Pseudocode for dTRPO

The pseudocode for dTRPO is shown in Algorithm 1.

## D. Experiment Details

### D.1. Training Pipeline

As illustrated in Figure 5, dTRPO follows a structured post-training workflow for dLLMs. Our base model, Fast-dLLM-v2-7B, is derived from Qwen2.5-7B-Instruct via diffusion pre-training. This initial training stage is functionally analogous to the Supervised Fine-Tuning (SFT) phase of Autoregressive Models (ARMs); however, notably, the dLLM is initialized from a well-trained ARM checkpoint rather than starting from scratch.

Subsequently, we execute the Direct Preference Optimization (DPO) stage leveraging our proposed method on the preference dataset. This phase serves as the counterpart to the offline Reinforcement Learning (RL) stage in traditional ARM pipelines, aimed at further refining the sequence probability distribution. The competitive baselines evaluated in Table 2 all focus on this specific alignment stage to ensure a fair and consistent comparison with dTRPO. We simply mix SmolTalk2, math, and code preference data evenly. Although the SmolTalk2 dataset is much larger than the other two, its data is less diverse and mostly consists of linguistic tasks requiring the model to write specific types of sentences. The benefit of mixing the data is shown in Table 4. We further show the results using 100k on-policy data points, where the prompts are from the OpenMathReasoning (Moshkov et al., 2025) and OpenCodeReasoning (Ahmad et al., 2025) datasets, and prompt the model to generate responses ahead of time. The on-policy data leads to better improvements in general instruction-following (IF) ability during DPO training. However, for rigorous tasks like math and coding, where it is difficult for the model to independently evolve quickly toward the correct logic, on-policy training falls short of the performance achieved using high-quality off-policy data.

### D.2. Generation Protocol

For LLaDA and Dream, we adopt the Fast-dLLM generation protocol enhanced with prefix caching (Wu et al., 2025), which we further adapt for batch generation[1]. For Fast-dLLM-v2, we utilize the official batch generation framework (Wu et al., 2026). Specifically, we employ greedy decoding with a threshold of $\epsilon = 1.0$, which corresponds to generating one token per

---

[1] https://github.com/NVlabs/Fast-dLLM

---

**Algorithm 1** dTRPO: Block-wise Ratio Reduction for Preference Optimization.

---

**Require:** Preference dataset $\mathcal{D}$ with triples $(x, y^+, y^-)$; policy model $\mu_\theta$; reference model $\mu_{\text{ref}}$; block count $N_B$; within-block steps $T_B$; scheduler $\text{Select}(\cdot)$ (e.g., top-$k$ by confidence); scale $\lambda$.

1: **for** each $(x, y^+, y^-) \in \mathcal{D}$ **do**
2:      Sample one time step per block: $t_s \sim U(1, T_B)$ for $s = 1, \ldots, N_B$
3:      **for** each completion $y \in \{y^+, y^-\}$ **do**
4:          Construct a packed, partially masked input $\tilde{\boldsymbol{\tau}}(y)$ using $\{t_s\}_{s=1}^{N_B}$ and the block-attention mask (Section B)
5:          Run one forward pass to obtain categorical predictions:

$$\mu_\theta(\cdot \mid \tilde{\boldsymbol{\tau}}(y)), \quad \mu_{\text{ref}}(\cdot \mid \tilde{\boldsymbol{\tau}}(y))$$

6:          For each block $s$, use the scheduler to select newly-unmasked indices

$$\mathcal{I}_s \leftarrow \text{Select}\big(\tilde{\boldsymbol{\tau}}(y), s\big)$$

7:          Compute the block-wise log-ratio score

$$S(y) \leftarrow \sum_{s=1}^{N_B} \sum_{i \in \mathcal{I}_s} \left[ \log \mu_\theta(y^{(i)} \mid \tilde{\boldsymbol{\tau}}(y)) - \log \mu_{\text{ref}}(y^{(i)} \mid \tilde{\boldsymbol{\tau}}(y)) \right]$$

8:      **end for**
9:      $z \leftarrow \lambda T_B \cdot \big(S(y^+) - S(y^-)\big)$
10:     $\mathcal{L}_{\text{dTRPO}} \leftarrow -\log \sigma(z)$
11: **end for**

---

*Table 4.* The benefit of different data source.

| Data Source | MATH | LCB | IF |
|---|---|---|---|
| Fast-dLLM-v2 | 60.26 | 59.1 | 62.11 |
| OpenMathReasoning + OpenCodeReasoning+On-policy Response | 62.74 | 59.10 | 65.62 |
| SmolTalk2 only | 63.18 | 13.36 | 67.10 |
| SmolTalk2 + Math + Code | 64.30 | 15.17 | 65.06 |

time step, supported by block-level caching[2].

For the additional evaluations involving the Qwen3 series, we leverage the LMDeploy (Contributors, 2023) framework as implemented by SDAR to facilitate efficient batch generation[3].

### D.3. Evaluation Dataset Details

All tasks are evaluated in a zero-shot setting, with the maximum generation length capped at 2048 tokens and the decoding temperature set to 0 to ensure deterministic output.

Specifically, we employ the `lm-eval-harness` framework (Gao et al., 2024)[4] to evaluate performance on GSM8K (Cobbe et al., 2021), GPQA (Rein et al., 2024), MATH (Hendrycks et al., 2021), and IFEval (Zhou et al., 2023). For coding benchmarks, we utilize the official LiveCodeBench evaluation protocol for LCBv6 (Jain et al., 2025)[5], while MBPP (Austin et al., 2021b) and HumanEval (Chen, 2021) are evaluated using the EvalPlus framework (Liu et al., 2023)[6].

---

[2]https://github.com/NVlabs/Fast-dLLM/tree/main/v2
[3]https://github.com/JetAstra/SDAR
[4]https://github.com/EleutherAI/lm-evaluation-harness
[5]https://github.com/LiveCodeBench/LiveCodeBench
[6]https://github.com/evalplus/evalplus

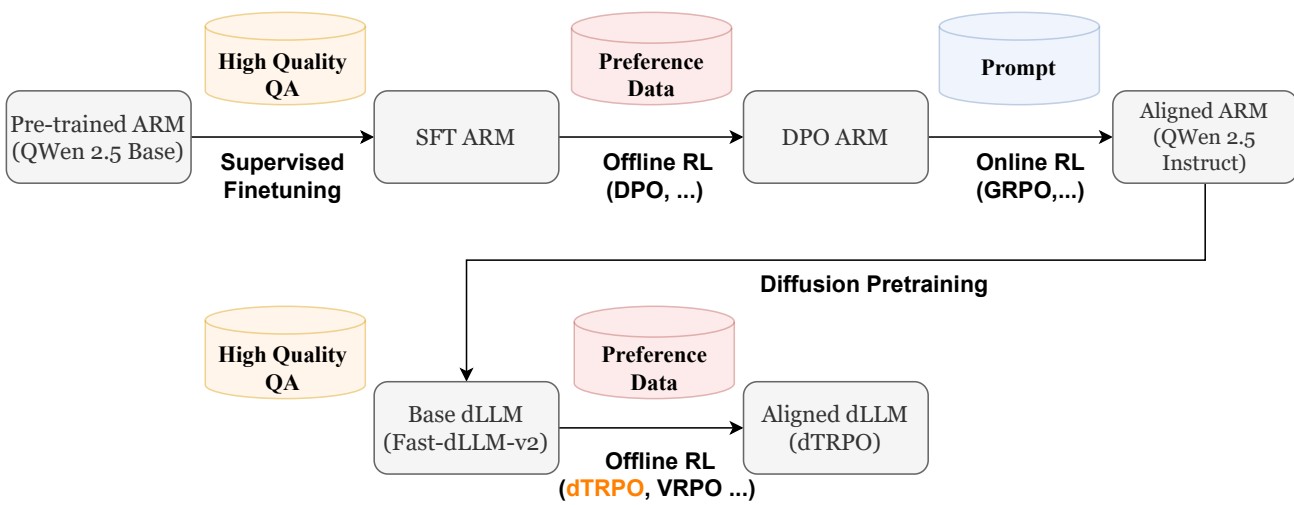

*Figure 5.* The post-training pipeline comparison between ARMs and dLLMs.

*Table 5.* Prompts used for different evaluation datasets.

| Role | Prompt Template |
|------|-----------------|
| *GSM8K (CoT):* | |
| **User** | Question: {question} |
| | Please reason step by step, and put your final answer within \boxed{ }. |
| *MATH:* | |
| **User** | Problem: {question} |
| | Please reason step by step, and put your final answer within \boxed{ }. |
| *GPQA (Generative, CoT, Diamond):* | |
| **User** | What is the correct answer to this question: {question} |
| | Choices: (A) {choice1} (B) {choice2} (C) {choice3} (D) {choice4} |
| | Let's think step by step: |
| *LiveCodeBench (v6):* | |
| **System** | You are a helpful assistant. |
| **User** | You will be given a question (problem specification) and will generate a correct Python program that matches the specification and passes all tests... |
| | Enclose your code within delimiters as follows: |
| | ```python |
| | # YOUR CODE HERE |
| | ``` |
| *MBPP+ & HumanEval+ (extra test):* | |
| **User** | Please provide a self-contained Python script... {Prompt} |
| **Assistant** | Below is a Python script with a self-contained function... |
| | ```python |

## D.4. Baseline Models and Comparison

We evaluate our method against state-of-the-art open-source dLLMs and several reproduced baselines built upon the `Fast-dLLM-v2-7B` architecture. The open-source dLLMs include LLaDA-Instruct (Nie et al., 2025)[7], LLaDA 1.5 (Zhu

---
[7]https://huggingface.co/GSAI-ML/LLaDA-7B-Instruct

et al., 2025)[8], Dream-Instruct (Ye et al., 2025)[9], and Fast-dLLM-v2 (Wu et al., 2026)[10]. For a fair comparison, all models are evaluated using our unified evaluation protocol.

To further validate our approach, we reproduce several training methodologies—including SFT, VRPO, and DPO with mean-field probability estimation—and adapt them for the block-wise dLLM framework. To ensure consistency, these baselines utilize the same training data and configurations as dTRPO. While the forward pass remains identical across these models, the primary distinction lies in the loss computation from the logits:

- **SFT**: Following Wu et al. (2026), we compute the loss for masked tokens using the ELBO-based objective (Ou et al., 2025).

- **VRPO**: The trajectory probability is derived from the ELBO over the masked tokens, which is subsequently optimized using the DPO objective.

- **DPO with Mean-field**: The trajectory probability is estimated by averaging the log-likelihoods over the masked tokens.

# E. More Comparisons

## E.1. Comparison in Other Backbone

*Table 6.* Performance Gain of our method with those dLLMs trained from QWen3.

| Model | GPQA | GSM8K | MATH | LCBv6 | MBPP+ | HumanEval+ | IFEval | Avg. Gain |
|---|---|---|---|---|---|---|---|---|
| Qwen3-8B | 51.52 | 89.61 | 82.98 | 42.18 | 66.40 | 80.50 | 82.99 | - |
| SDAR | 28.28 (–23.24) | 91.28 (+1.67) | 74.22 (–8.76) | 20.57 (–21.61) | 66.40 (0.0) | 65.90 (–14.6) | 63.03 (–19.96) | -12.36 |
| TraDO | 38.89 (–12.63) | 91.05 (+1.44) | 78.74 (–4.24) | 29.00 (–13.1) | 66.40 (0.0) | 67.10 (–13.4) | 62.85 (–20.14) | -8.88 |
| Qwen2.5-7B | 36.36 | 87.87 | 73.06 | 24.42 | 67.50 | 74.40 | 71.38 | - |
| dTRPO | 30.30 (–6.06) | 85.97 (-1.90) | 64.30 (–8.76) | 15.17 (–9.25) | 51.60 (–15.9) | 63.40 (–11.0) | 65.06 (–6.32) | -8.46 |

**Qwen3** A key motivation for diffusion LLMs is to inherit capabilities from strong ARMs through initialization and then perform post-training under the diffusion policy, as illustrated in Figure 5. We further compare our method with SDAR (Cheng et al., 2025) and TraDO (Wang et al., 2025b), two state-of-the-art dLLMs initialized from Qwen3 (Yang et al., 2025) in Table 6. Here we omit benchmarks evaluated via LLM-as-a-judge.

SDAR and TraDO exhibit less stable adaptation: they can be strong on some benchmarks (e.g., MBPP, GSM8K) but degrade substantially on others (e.g., GPQA, IFEval). In contrast, our method achieves a more consistent performance profile. Importantly, as discussed in Figure 1 and Section 4.3, our training is purely offline, does not require on-policy generation, and does not rely on benchmark-specific training sets.

**LLaDA** We report results in Table 7 using the dPad (Chen et al., 2025b) inference protocol with dual cache and a shorter generation length, noted in the brackets.

*Table 7.* LLaDA with post-training methods.

| Method | MATH (256) | LCB (1024) | IF (512) |
|---|---|---|---|
| LLaDA | 27.84 | 10.29 | 55.08 |
| + ELBO-SFT | 30.71 | 7.43 | 59.15 |
| + VRPO | 29.08 | 6.16 | 56.93 |
| + dTRPO | 34.88 | 10.86 | 61.00 |

**1B Scale Model.** We report the post-training performance based on Fast-dLLM-v2 1.5B[11] in Table 8.

---

[8] https://huggingface.co/GSAI-ML/LLaDA-1.5
[9] https://huggingface.co/Dream-org/Dream-v0-Instruct-7B
[10] https://huggingface.co/Efficient-Large-Model/Fast_dLLM_v2_7B
[11] https://huggingface.co/Efficient-Large-Model/Fast_dLLM_v2_1.5B

*Table 8.* Fast-dLLM-v2 1.5B with post training methods.

| Method | MATH | LCB | IF |
|---|---|---|---|
| Fast-dLLM-v2 1.5B | 37.9 | 5.4 | 44.73 |
| + ELBO-SFT | 34.86 | 5.31 | 35.3 |
| + VRPO | 39.66 | 5.69 | 43.81 |
| + dTRPO | 38.96 | 6.45 | 44.36 |

## E.2. Ablation on the Parameter Efficient Finetuning

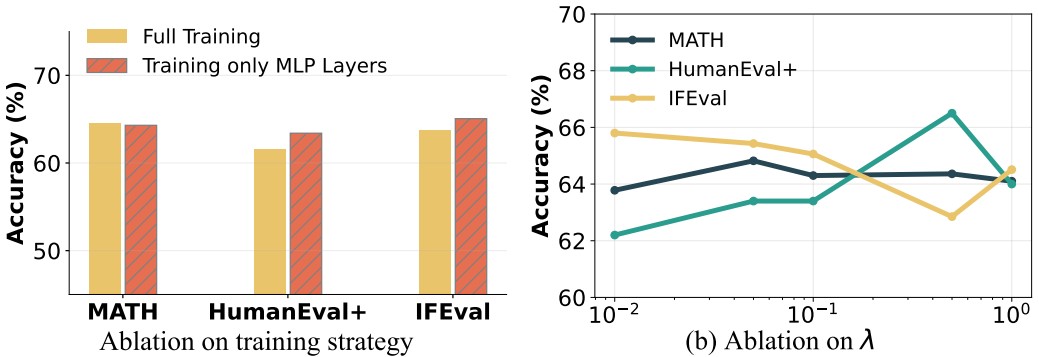

*Figure 6.* Ablation on the parameter efficient finetuning and DPO hyperparameters.

In our training setup, we selectively update only the MLP layers and the output projection layers. Given that Fast-dLLM-v2 is initialized from the Qwen2.5, such parameter-efficient training is crucial for preserving the pre-trained knowledge and preventing catastrophic forgetting. As illustrated in Figure 6(a), restricting the optimization to MLP layers not only significantly reduces the computational overhead but also leads to superior performance across benchmarks compared to full-parameter fine-tuning.

The most critical hyperparameter in our DPO-style objective is the KL-divergence regularization coefficient, $\lambda$, which balances the sequence probability maximization and the divergence from the reference model. We conduct an ablation on $\lambda$ in Figure 6(b). The results show that the model performance remains stable across a wide range of $\lambda$ values, indicating that our proposed method is robust and not overly sensitive to the precise choice of regularization strength.

# F. Qualitative Analysis

We provide some qualitative examples of dTRPO and Qwen2.5-Instruct on coding and instruction following tasks. From our observations, dTRPO can generate correct answers with faster speed and follow the instructions more closely.

---

**Math Question**

Jen enters a lottery by picking 4 distinct numbers from $S = \{1, 2, 3, \cdots, 9, 10\}$. 4 numbers are randomly chosen from $S$. She wins a prize if at least two of her numbers were 2 of the randomly chosen numbers, and wins the grand prize if all four of her numbers were the randomly chosen numbers. The probability of her winning the grand prize given that she won a prize is $tfracmn$ where $m$ and $n$ are relatively prime positive integers. Find $m + n$.

*Qwen2.5-Instruct: 992 tokens, 137.00s*

To solve this problem, we need to calculate the probability of Jen winning the grand prize given that she has won a prize. We will use conditional probability for this purpose.

---

First, let's define the events: - $A$: Jen wins the grand prize. - $B$: Jen wins a prize (at least 2 of her numbers match the 4 randomly chosen numbers).

We need to find $P(A \mid B)$, which is the probability of Jen winning the grand prize given that she has won a prize. By the definition of conditional probability, we have:

$$P(A \mid B) = \frac{P(A \cap B)}{P(B)}.$$

Since event $A$ (winning the grand prize) is a subset of event $B$ (winning a prize), $A \cap B = A$. Therefore, we have:

$$P(A \mid B) = \frac{P(A)}{P(B)}.$$

Now, let's calculate $P(A)$ and $P(B)$.
Step 1: Calculate $P(A)$ The probability of Jen winning the grand prize is the probability that all 4 of her chosen numbers match the 4 randomly chosen numbers. Since there is only one way to choose 4 specific numbers out of 10, and Jen's choice must be exactly these 4 numbers, we have:

$$P(A) = \frac{1}{\binom{10}{4}} = \frac{1}{210}.$$

Step 2: Calculate $P(B)$ The probability of Jen winning a prize means that at least 2 of her chosen numbers match the 4 randomly chosen numbers. We can use complementary counting to find this probability. First, we calculate the probability of the complementary event, which is that Jen wins no prize or wins exactly 1 prize.
Complementary Event 1: No Prize The number of ways to choose 4 numbers such that none of Jen's 4 numbers are included is:

$$\binom{6}{4} = 15.$$

So, the probability of this event is:

$$\frac{\binom{6}{4}}{\binom{10}{4}} = \frac{15}{210} = \frac{1}{14}.$$

Complementary Event 2: Exactly 1 Prize The number of ways to choose 4 numbers such that exactly 1 of Jen's 4 numbers is included is:

$$\binom{4}{1} \cdot \binom{6}{3} = 4 \cdot 20 = 80.$$

So, the probability of this event is:

$$\frac{80}{210} = \frac{8}{21}.$$

Therefore, the probability of the complementary event (no prize or exactly 1 prize) is:

$$\frac{1}{14} + \frac{8}{21} = \frac{3}{42} + \frac{16}{42} = \frac{19}{42}.$$

Thus, the probability of Jen winning a prize is:

$$P(B) = 1 - \frac{19}{42} = \frac{23}{42}.$$

Step 3: Calculate $P(A \mid B)$ Now we can find the conditional probability:

$$P(A \mid B) = \frac{P(A)}{P(B)} = \frac{\frac{1}{210}}{\frac{23}{42}} = \frac{1}{210} \cdot \frac{42}{23} = \frac{42}{4830} = \frac{1}{115.5} = \frac{1}{115.5} = \frac{1}{115.5} = \frac{1}{115.5} = \dots$$

*dTRPO: 422 tokens, 13.53s*

To determine the probability that Jen wins the grand prize given that she won a prize, we need to use conditional probability. Let's define the events as follows: - $A$: Jen wins a prize. - $B$: Jen wins the grand prize.

We need to find $P(B \mid A)$, the probability that Jen wins the grand prize given that she won a prize. By the definition of conditional probability, we have:

$$P(B \mid A) = \frac{P(A \cap B)}{P(A)}$$

First, we calculate $P(A)$, the probability that Jen wins a prize. Jen wins a prize if at least two of her numbers match the randomly chosen numbers. We will use the complementary probability approach to find $P(A)$. This involves calculating the probability that Jen has fewer than 2 matches and subtracting it from 1.

1. **Calculate the total number of ways to choose 4 numbers from $S$:**

$$\binom{10}{4} = \frac{10!}{4!6!} = 210$$

2. **Calculate the number of ways to choose fewer than 2 matches:** - **Case 1: 0 matches** Jen chooses 4 numbers from the 6 numbers not chosen by the lottery.

$$\binom{6}{4} = \binom{6}{2} = 15$$

- **Case 2: 1 match** Jen chooses 1 match from the 4 numbers chosen by the lottery and 3 non-matches from the 6 numbers not chosen.

$$\binom{4}{1} \times \binom{6}{3} = 4 \times 20 = 80$$

Adding these cases together gives the total number of ways to choose fewer than 2 matches:

$$15 + 80 = 95$$

Therefore, the number of ways to choose at least 2 matches is:

$$210 - 95 = 115$$

The probability that Jen wins a prize is:

$$P(A) = \frac{115}{210} = \frac{23}{42}$$

3. **Calculate $P(A \cap B)$, the probability that Jen wins the grand prize:** Jen wins the grand prize if all 4 numbers she picks match the 4 numbers chosen by the lottery. There is only 1 way for this this:

$$P(A \cap B) = \frac{1}{210}$$

4. **Calculate $P(B \mid A)$:**

$$P(B \mid A) = \frac{P(A \cap B)}{P(A)} = \frac{\frac{1}{210}}{\frac{23}{42}} = \frac{1}{210} \times \frac{42}{23} = \frac{42}{4830} = \frac{1}{115}$$

The probability that Jen wins the grand prize given that she won a prize is $\frac{1}{115}$. Since $m = 1$ and $n = 115$, we have $m + n = 1 + 115 = 116$.

Thus, the final answer is:

$$\boxed{116}$$

**Coding Question**

fix the error in this prgram in js

```
<p>Write a program to find the largest number among 3 numbers.</p>
<input type=text placeholder=Enter 1st number id=t1>

<input type=text placeholder=Enter 2nd number id=t2>

<input type=text placeholder=Enter 3rd number id=t3>
<button onclick=check()>Check</button>
<h3 id=ans>The largest number is</h3>
<script>
    function check(){
        let n1 = document.getElementById( t1 ).value;
        let n2 =document.getElementById(t2).value;
        let n3 = document.getAnimations(t3).value;

        if (n1>n2 && n1>n3) {
            document.getElementById( ans ).innerHTML =The largest is +num1;
        } else if (n2 > n3) {
            document.getElementById( ans ).innerHTML =The largest is  +num2;
        }else{
            document.getElementById(ans).innerHTML = The largest is + num3;
        }
    }
}
</script>
```

*Qwen2.5-Instruct: 187 tokens, 11.65s*

There are several issues with your JavaScript code that need to be fixed:

- 1. The `getAnimations` method does not exist in JavaScript. It should be **`getElementById`**.

- 2. The variables `num1`, `num2`, and `num3` are used instead of `n1`, `n2`, and `n3`.

- 3. The comparison logic can be simplified.

Here is the corrected version of your code:

```
<p>Write a program to find the largest number among 3 numbers.</p>
<input type=text placeholder=Enter 1st number id=t1>

<input type=text placeholder=Enter 2nd number id=t2>

<input type=text placeholder=Enter 3rd number id=t3>
<button onclick=check()>Check</button>
<h3 id=ans>The largest number is</h3>
<script>
    function check() {
        let n1 = parseFloat(document.getElementById(t1).value);
        let n2 = parseFloat(document.getElementById(t2).value);
        let n3 = parseFloat(document.getElementById(t3).value);

        let largest = Math.max(n1, n2, n3);

        document.getElementById(ans).innerHTML = The largest is  + largest;
    }
</script>
```

**Explanation:**

1. **Corrected Method Name**: Changed `getAnimations` to `getElementById`.

2. **Variable Names**: Used `n1`, `n2`, and `n3` instead of `num1`, `num2`, and `num3`.

3. **Simplified Logic**: Used **`Math.max`** to find the largest number among the three inputs.

4. **Parsed Input as Float**: Used **`parseFloat`** to convert the input values from strings to numbers.

This should work correctly now.

---

*dTRPO: 195 tokens, 6.75s*

The error in your JavaScript code is in the line where you're trying to get the value of the third input field. Instead of `document.getAnimations("t3").value`, you should use **`document.getElementById("t3").value`**. Also, you need to declare `num1`, `num2`, and `num3` as variables before using them in the comparison. Here's the corrected code:

```
<p>Write a program to find the largest number among 3 numbers.</p>
<input type=text placeholder=Enter 1st number id=t1>

<input type=text placeholder=Enter 2nd number id=t2>

<input type=text placeholder=Enter 3rd number id=t3>
<button onclick=check()>Check</button>
<h3 id=ans>The largest number is</h3>

<script>
   function check() {
       let n1 = document.getElementById(t1).value;
       let n2 = document.getElementById(t2).value;
       let n3 = document.getElementById(t3).value;

       // Convert inputs to numbers
       n1 = Number(n1);
       n2 = Number(n2);
       n3 = Number(n3);

       if (n1 > n2 && n1 > n3) {
          document.getElementById(ans).innerHTML = The largest is  + n1;
       } else if (n2 > n3) {
          document.getElementById(ans).innerHTML = The largest is  + n2;
       } else {
          document.getElementById(ans).innerHTML = The largest is  + n3;
       }
   }
</script>
```

Additionally, I added **`Number()`** to convert the input values to numbers before comparing them, as the input values are initially strings. This will ensure that the comparison works correctly.

---

**Instruction Following Question**

write an upbeat scifi short story

---

*Qwen2.5-Instruct: 817 tokens, 34.06s*

The Last Frontier
In the year 2145, humanity had spread across the stars, but the final frontier remained: the uncharted planet of Zephyria. It was a world of perpetual twilight, with vast oceans that shimmered like liquid silver under the dim light of its twin suns. The planet's atmosphere was rich in oxygen and nitrogen, making it a prime candidate for terraforming. However, the harsh conditions and unknown dangers had deterred all previous attempts.

Captain Elara Myles stood at the helm of the starship *Aurora*, her eyes fixed on the holographic map of Zephyria. She had been chosen to lead the first expedition to Zephyria, a mission that could either mark the beginning of a new era or end in disaster. Her crew consisted of the best and brightest from Earth's most advanced space agencies—engineers, scientists, and explorers.

As they approached the planet, the *Aurora* encountered a strange phenomenon. A series of lights began to dance around them, pulsating in a pattern that seemed almost... intelligent. Elara's heart raced as she ordered the ship to slow down. The lights grew closer, forming a shape that resembled a giant eye, its iris a swirling mass of colors.

"Captain, we're receiving a signal," said Lieutenant Kael, the communications officer, his voice tinged with excitement.

Elara nodded, her fingers dancing over the control panel. "Acknowledge the signal and prepare to receive data."

The ship's systems processed the incoming information, and a detailed map of Zephyria appeared on the main screen. But there was something else—a message, written in a language that none of them could understand. It was a warning, a plea for help, and a promise of knowledge.

"Captain, we need to land," said Dr. Lena Chen, the chief scientist, her voice filled with urgency. "This is our chance to make contact with whatever is out there."

Elara hesitated for a moment, then gave the order. The *Aurora* descended through the clouds, its engines humming softly as it touched down on a vast, open plain. The crew stepped out onto the alien soil, their boots sinking slightly into the soft, nutrient-rich ground.

As they set up their base camp, the lights reappeared, this time forming a structure that looked like a giant dome. The crew gathered around it, their hearts pounding with anticipation.

"Are you ready?" Elara asked, her voice steady despite the thrill of the moment.

"Always, Captain," replied Kael, his eyes gleaming with determination.

With a collective nod, the team approached the dome. As they reached it, the structure opened, revealing a chamber filled with bioluminescent plants and floating crystals that emitted a soothing light. In the center of the room, a figure materialized—a being of pure energy, its form shifting and changing before their eyes.

"Welcome, travelers," the entity said, its voice resonating through the chamber. "I am Zephyrion, guardian of this world. You have come at a time of great need. We have faced threats that even your advanced technology cannot comprehend. We seek allies who can help us protect our home."

Elara stepped forward, her hand outstretched. "We are here to learn and to help. What can we do?"

Zephyrion smiled, its form stabilizing into a humanoid shape. "Together, we can unlock the secrets of Zephyria and perhaps find a way to bridge the gap between our worlds. But first, you must prove your worth."

With that, Zephyrion led the team through a series of challenges, each one testing their skills and resolve. They learned to harness the planet's unique energy sources, discovered new forms of life, and even uncovered hidden treasures that could change the course of human history.

By the time the *Aurora* prepared to leave, the crew had formed a deep bond with their new allies. Zephyrion had given them a gift—a device that would allow them to communicate with the planet's inhabitants and share their knowledge.

As they lifted off, Elara turned to her crew, her face alight with hope. "We've just begun. There's so much more to explore, to learn, and to share."

Kael grinned, his eyes shining. "And we'll do it together, Captain."

The *Aurora* soared into the sky, leaving behind a world that had once seemed unreachable. But now, with the promise of a new alliance and the potential for endless discovery, Zephyria was no longer a distant dream—it was a new beginning.

—

The journey back to Earth was filled with wonder and excitement. The crew shared their experiences with the world, inspiring a new generation of explorers and scientists. And though Zephyria remained a distant world, its spirit lived on in the hearts of those who had answered the call.

For in the vast expanse of the universe, there were always new frontiers to conquer, new mysteries to uncover, and new friends to make. And sometimes, the greatest discoveries were not found in the stars, but in the bonds we forge along the way.

*dTRPO:* *338 tokens, 7.02s*

Title: The Quantum Quants and the Quantum Quirk

In the bustling city of Harmonia, where science and art intertwine, a group of quantum physicists, known as the "Quantum Quants," had just made a groundbreaking discovery. Led by the brilliant and enthusiastic Dr. Ada Quantum, their team had developed a new technology that could manipulate quantum entanglement for practical applications.

One day, while conducting an experiment in their quantum computing lab, they stumbled upon an unexpected phenomenon. A minor glitch in their quantum computer caused particles to become entangled in a way that defied all known physics. The team dubbed this peculiar occurrence the "Quantum Quirk."

Excited by the possibilities, the Quantum Quants decided to harness this newfound power for good. They devised a device, the "Quantum Compass," which could teleport small objects across vast distances with unparalleled precision. The implications were staggering - instantaneous communication, global transportation, and even the potential to solve some of the world's most pressing problems.

The Quantum Quants embarked on a mission to share their invention with the world. They traveled to cities across Harmonia, demonstrating the Quantum Compass's capabilities and sparking a wave of optimism and innovation. People were amazed as they watched objects teleport through their fingertips, and the potential applications were endless.

However, the Quantum Quants faced unexpected challenges. Some governments and corporations sought to control the technology for their own gain, while others feared the potential misuse of such advanced power. But the team remained steadfast in their commitment to ethical use. They established a global network of quantum researchers and engineers, ensuring that the Quantum Compass was used responsibly and equitably.

As the Quantum Quants continued to spread their message of unity and progress, the world became a brighter, more connected place. The Quantum Compass brought people together, bridging gaps between nations and fostering a sense of shared humanity. And all thanks to a quantum glitch, the Quantum Quirk, that had once been a minor setback became the catalyst for a new era of scientific exploration and cooperation.

