# OpenReview forum: "dTRPO : Trajectory Reduction in Policy Optimization of Diffusion Large Language Models"
_ICML.cc/2026/Conference — ICML 2026 regular_

### Official Review · Reviewer_n67A · 2026-03-07

**Soundness:** 3
**Presentation:** 3
**Significance:** 3
**Originality:** 3
**Overall Recommendation:** 5
**Confidence:** 4

**Summary:**

In this submission, the authors investigate how to aligning difussion Large Language Models (dLLMs) with human preference. This paper finds a limitation in existing alignment method for dLLMs: Compared with the token-factorized probabilities of autoregressive-based LLMs, trajectory probability is much more computationally expensive and unstable to compute for dLLMs. To solve this issue, the authors propose a framework, Trajectory Reduction Policy Optimization (dTRPO), to calculuate the probabilities more efficiently by designing an estimator that uses both state reduction and ratio reduction. Experimental results on 7B-scale dLLMs validate the proposed dTRPO is effective in improving the performance of dLLMs such as reasoning and instruction-following ability.

**Compliance With Llm Reviewing Policy:**

Affirmed.

**Final Justification:**

Overall, this paper identifies a limitation in existing alignment methods for diffusion Large Language Models and proposes a new framework, Trajectory Reduction Policy Optimization (dTRPO), to compute probabilities more efficiently by introducing an estimator that incorporates both state reduction and ratio reduction.

I think this work is novel and the provided proof are also rigirous. Extensive experimental results also validate its effectiveness. Originally, I am concerned about whether this work can be transferred to PPO-style training and its scalability to different models. During the rebuttal period, the authors first conducted an ablation study focusing specifically on the on-policy aspect of such training which addressed my concern about PPO-style training.  Additionally, the new results on 1B-scale model addressed my concern about scalability.

Therefore, my final recommendation is Accept.

**Key Questions For Authors:**

I have the folowing questiosn/suggestions:
- It would be beneficial to conduct experiments on more models with different sizes and model architectures, especially for training frameworks.
- I am wondering if it is possible to conduct some experments based on PPO.
- In the training details part, the $\lambda$ is set to 0.05. I am curious the impact of this hyperparameter.

**Limitations:**

There is not a seciton describing the limitations.

**Strengths And Weaknesses:**

This submission has the following **strengths**:
- This paper is well-organized which makes it easy to read.
- It is important to investigate how to enhance the generation of diffusion-based Large Language Models (dLLMs) which is the research problem of this work.
- The proofs in this paper are comprehensive and provide a theoritical foundation.
- Experiments on datasets like GPQA and GSM8K validate the effetiveness of the proposed method in improving the performance of 7B dLLMs.

This submission has the following **weaknesses**:
- The experiments are conducted only on 7B-scale dLLMs. It is unclear whether the proposed reinforcement learning framework works for smaller or larger dLLMs. The used model architecture is also limited.
- Although this submission provides theoretical analysis for PPO-style surrogates. However, the experiments are based only on DPO.

There are some typos:
- Appendix C and Appendix D share the same name. It appears to be a duplicate.
- In page 5, there is no '.' in this sentence 'as the first row in Table 1'.

---

> ### Author Rebuttal · Authors · 2026-03-30
>
> We appreciate the reviewer's recognition of our presentation, the algorithm and the experiment, and we will fix the typos according to the reviewer's comment. Below is our response
>
> ### **The experiments are conducted only on 7B-scale dLLMs. It is unclear whether the proposed reinforcement learning framework works for smaller or larger dLLMs. The used model architecture is also limited.**
>
>
> We agree that evaluating primarily on the Fast-dLLM-v2 7B model makes the initial experimental scope seem somewhat narrow. To clarify our choice of architectures: at the time of conducting the main experiments, Fast-dLLM-v2 and LLaDA were practically the only publicly available models with robust infrastructure support (e.g., well-implemented block attention, caching, checkpointing, and training scripts). This infrastructure is essential for scaling up dLLM post-training efficiently.
>
>
> Nevertheless, we completely agree that diversifying the evaluation across different scales and architectures strengthens our claims. Therefore, we have conducted additional experiments on the LLaDA architecture (expanding on the brief results in Figure 3c) and the 1B-scale Fast-dLLM-v2 model.
>
> Due to time constraints during the rebuttal, we report the following LLaDA results using the dPad inference protocol with dual cache and a shorter generation length (noted in brackets). (Note: In the original paper, we reported performance using the lm-eval protocol with a generation length of 2048 and without caching).
>
> | Method | MATH (256) | LCB (1024) | IF (512) |
> | :--- | :--- | :--- | :--- |
> | **LLaDA** | 27.84 |10.29| 55.08|
> | **ELBO-SFT** | 30.71| 7.43| 59.15|
> | **VRPO** | 29.08 | 6.16 |56.93|
> | **dTRPO**|34.88|10.86 | 61|
>
>
> Below is Fast-dLLM-v2 1B performance
>
> | Method | MATH  | LCB  | IF|
> | :--- | :--- | :--- | :--- |
> | **Fast-dLLM-v2** | 37.9 |5.4| 44.73 |
> | **ELBO-SFT** | 34.86| 5.31| 35.3|
> | **VRPO** | 39.66 |5.69 |43.81|
> | **dTRPO**|38.96|6.45 | 44.36|
>
>
> We observe that different architectures exhibit distinct behaviors when applying various post-training methods. For example, on LLaDA, training with ELBO-SFT yields a relatively better effect than VRPO, whereas on the 1B model, VRPO shows slightly better performance in math. Moreover, improper training objectives can have a high negative impact on LLaDA, and the overall headroom for improvement on the smaller 1B model is naturally more limited. Overall, however, our proposed dTRPO method consistently achieves SOTA or highly comparable performance across these different architectures and scales.
>
>
> ### **I am wondering if it is possible to conduct some experments based on PPO.**
> We agree that exploring PPO-style training is an important next step. Transitioning from DPO to PPO introduces two main complexities for dLLMs: optimizing a PPO surrogate objective and utilizing on-policy (online) generation. Given the short rebuttal window, it is difficult to implement and tune a full, stable PPO pipeline from scratch.
>
>
> However, to partially address your suggestion, we ran an ablation focusing specifically on the on-policy aspect of such training. We used 100k on-policy data points generated by the model itself (using prompts from the OpenMathReasoning and OpenCodeReasoning datasets). For the preference pairs, we treated the original dataset's ground-truth response as the preferred response, and the model's generated response as the dispreferred one. We then trained the model using our framework. The results are as follows:
>
> | Data | MATH | HumanEval+ | IF |
> | :--- | :--- | :--- | :--- |
> | **Baseline (Fast-dLLM-v2)** | 60.26 |  59.1 | 62.11|
> | **Off-policy** |64.3 |63.4 | 65.06|
> | **On-policy** | 62.74 |59.10 | 65.62|
>
> These results align with our expectations. Because the on-policy data is generated by the model itself, the distribution gap is smaller compared to external off-policy data. This naturally leads to stronger improvements in general instruction-following (IF) ability. However, for rigorous tasks like math and coding, where it is difficult for the model to independently evolve toward the correct logic without external guidance, on-policy training falls short of the performance achieved using high-quality off-policy data.
>
> ### **In the training details part, the $\beta$ is set to 0.05. I am curious the impact of this hyperparameter.**
> Please refer to Figure 5(b) in the paper, where we evaluate the model's performance across different values.

---

> > ### Author Rebuttal · Reviewer_n67A · 2026-04-02
> >
> > I thank the authors for their detailed and clarifying rebuttal. My concerns are fully addressed. For my Q2 about PPO-style training, I understand that it is hard to provide a more comprehensive result during the rebuttal period. But I hope that additional results can be included in the next revision. I have increased my evaluation score accordingly.

---

### Official Review · Reviewer_w4Wk · 2026-03-11

**Soundness:** 2
**Presentation:** 2
**Significance:** 1
**Originality:** 2
**Overall Recommendation:** 4
**Confidence:** 3

**Summary:**

Diffusion Large Language Models pose new challenges for human preference alignment. This work optimizes the dLLM generation process by theoretically formulating and efficiently quantifying the probability of generation trajectories. We prove two properties: (i) under reference policy regularization, the probability ratio of intermediate diffusion states equals that of newly unmasked tokens, and (ii) the total generation probability is estimable via a single forward pass using block attention. Integrating these estimates into preference optimization yields dTRPO. Evaluated on 7B dLLMs, dTRPO substantially outperforms state-of-the-art baselines in instruction-following and reasoning benchmarks, achieving gains of up to 9.6% (STEM), 4.3% (coding), and 3.0% (instruction-following).

**Compliance With Llm Reviewing Policy:**

Affirmed.

**Final Justification:**

After examining the appendix and codes, I slightly changed my mind to go toward weak acceptance.

**Key Questions For Authors:**

The paper proposes that the probability ratio of intermediate diffusion states is equivalent to the probability ratio of newly unmasked tokens. This is an interesting and potentially important insight. However, it would be helpful if the authors could clarify the following points: (1) Under what assumptions does this equivalence hold? (2) Is the relationship exact, or does it rely on certain approximations in practice? (3) Does this interpretation have implications for designing new decoding or training strategies beyond the current method? Further clarification on the theoretical scope of this proposition would help readers better understand its significance.

The design of the proposed method appears to be particularly compatible with structured reasoning tasks such as mathematical problems, where token dependencies and iterative refinement may be important. However, this motivation is not explicitly discussed in the paper. Was the method intentionally designed with such tasks in mind? If so, could the authors elaborate on the design intuition behind this choice? Have the authors observed particular advantages of the method in reasoning-heavy tasks such as mathematics? Providing more discussion on this point could help clarify the intended strengths of the approach.

In Table 2, the evaluation mainly compares the proposed method with Qwen and LLaDA. Could the authors comment on why these particular baselines were selected? Are there plans to evaluate the method against a broader set of models? A wider set of baselines might help readers better understand the relative position of the proposed method. In addition, it might improve clarity if the best performance across all models (including baselines) were clearly highlighted in the table.

According to Table 2, the proposed method improves upon LLaDA and approaches the performance of Qwen2.5 Instruct, but there still appears to be a noticeable gap in performance. Could the authors provide insight into what factors they believe are responsible for this remaining gap? Do the authors see the proposed approach primarily as an incremental step toward bridging this gap? Understanding this point would help clarify the practical implications of the method.

Related to the previous point, current Qwen2.5 Instruct-style architectures appear to achieve significantly stronger performance in many tasks. The proposed method seems to represent one possible step toward improving diffusion-style generation methods. However, it would be interesting to hear the authors’ perspective on broader architectural directions. Do the authors envision integrating the proposed approach more deeply with architectures similar to Qwen2.5 Instruct? Are there more radical or alternative design directions that the authors believe could further close the performance gap with strong autoregressive models? What aspects of the current architecture do the authors believe most limit the achievable performance? The authors’ insights on these potential future directions would be valuable for understanding how this line of work might evolve.

**Limitations:**

No explicit limitation section is set up.

Given the design guidelines, I don't think it would improve performance by uniformly applying it to general NLP downstream tasks.
I think it's fine to describe this as a limitation. What do you think?

**Strengths And Weaknesses:**

Strengths
Soundness: The paper presents a theoretically motivated insight that connects probability ratios in diffusion models with autoregressive generation mechanisms. Specifically, the paper claims that the probability ratio of intermediate diffusion states is equivalent to the probability ratio of newly unmasked tokens. This proposition is conceptually interesting and appears to be a reasonable interpretation of the relationship between the diffusion process and token generation. The idea provides a bridge between two different perspectives on language generation: diffusion-based modeling and autoregressive token prediction. Such a connection may help clarify how token probabilities can be interpreted during intermediate diffusion steps. From a theoretical standpoint, this proposition is intriguing and appears internally consistent within the framework described in the paper.

Significance: The work attempts to contribute to the broader understanding of how diffusion-based generative models can relate to more traditional autoregressive language generation mechanisms. Establishing such conceptual connections can be valuable for understanding the theoretical properties of generative models and for guiding the design of new algorithms. In addition, the empirical results indicate that the method achieves competitive performance on several downstream tasks such as instruction-following and coding tasks, suggesting that the approach is not restricted to a single specialized domain. This indicates that the method may have a reasonably broad range of applicability.

Originality: The main originality of the paper lies in the proposed interpretation of the relationship between intermediate diffusion states and token-level probability ratios. The idea that these probability ratios correspond to the probabilities of newly unmasked tokens offers a new conceptual perspective on how diffusion-based text generation may relate to sequential token generation. Even if the individual components involved in the framework may be related to existing modeling paradigms, the way they are connected in the paper represents a creative attempt to reinterpret the behavior of diffusion models in terms of token-level probabilistic reasoning.

Presentation: The central idea of the paper-the equivalence between the probability ratios of intermediate diffusion states and newly unmasked tokens-is relatively easy to grasp and forms a clear conceptual core for the work. The experimental section provides comparisons with existing models and presents results in a tabular format, which helps readers understand the empirical performance of the proposed approach. Overall, the paper communicates the basic intuition behind the method in a reasonably clear way.

Weaknesses
According to Table 2, the proposed method improves upon LLaDA, and its performance approaches that of Qwen2.5 Instruct. However, it still appears to fall somewhat short of the performance of Qwen2.5 Instruct. Given that Qwen2.5 Instruct represents a strong baseline, this gap raises questions about the practical competitiveness of the proposed approach. In particular, if recent improvements to Qwen-based models have already achieved substantially stronger results, the proposed method may still remain significantly behind the current state of the art in terms of empirical performance. Therefore, while the method shows improvement over some diffusion-based baselines, the overall empirical advantage of the approach remains somewhat limited.

The experimental comparison appears somewhat narrow. In Table 2, the primary baselines are Qwen and LLaDA, but additional baselines could strengthen the evaluation. Including more models would make it easier to understand the relative strengths and weaknesses of the proposed method. In particular, comparisons with a wider range of models could provide a clearer picture of how competitive the approach is within the broader ecosystem of language models. In addition, the formatting of Table 2 could be improved. The best performance across all models, including both the proposed method and baselines such as Qwen and LLaDA, should be clearly highlighted (e.g., in bold) so that readers can easily identify the strongest results.

Although the method appears to have characteristics that may make it particularly suitable for certain types of tasks (for example, tasks that require structured reasoning such as mathematics), this design motivation is not explicitly discussed in the paper. The paper would benefit from a clearer explanation of why the proposed formulation is expected to work well for certain types of problems, and what kinds of tasks the method is particularly suited for. Without this clarification, some aspects of the method's design appear implicit rather than fully motivated.

While the central theoretical proposition is interesting, its implications for practical model behavior are not fully explored in the paper. In particular, it would be helpful if the authors provided more explicit discussion of how the proposed probability-ratio perspective influences the design of the algorithm and why it leads to the observed empirical results. Expanding this explanation could improve the clarity of the contribution and help readers better understand the significance of the theoretical insight.

---

> ### Author Rebuttal · Authors · 2026-03-30
>
> We sincerely thank the reviewer for recognizing the theoretical value and empirical potential of our work. The primary goal of our paper is to advance the alignment of dLLMs by providing both a rigorous theoretical foundation and a highly practical algorithm. By fundamentally improving how dLLMs are aligned, we significantly close the performance gap with ARMs across various backbones(Table 2, Response to reviewer n67A). Successfully closing this performance gap unlocks massive generation efficiency benefits (16.2 $\to$ 29.87; Please see the second question of reviewer DbBn).
> In practical production environments, the industry and research field is actively pushing this direction (https://deepmind.google/models/gemini-diffusion/) because multi-token parallel decoding can resolve straightforward queries up to 100x faster than standard ARMs.
>
> ### **it is particularly suitable for certain types of tasks**
> Our method was not intentionally designed with these specific tasks in mind. Our explicit goal, as stated in the introduction, is "aligning dLLMs for human preference" (Line 15) and "equipping models with strong zero-shot instruction-following ability" (Line 42). In standard RLHF, a reward model evaluates responses, while DPO simplifies this via a contrastive learning objective. However, applying DPO to dLLMs faces a major obstacle: computing the required trajectory probabilities during the forward pass is computationally prohibitive compared to ARMs (Line 52).
>
> ### **how the proposed probability-ratio perspective influences the design of the algorithm and why it leads to the observed empirical results**
> In standard alignment, a reward model evaluates how well a response follows a user's instruction, and the algorithms maximize the reward. DPO simplifies this by applying contrastive learning objective with preference dataset an shows that this optimization is equivalent to maximizing the reward in RLHF.
> Our paper solves exact bottleneck of the probability computation of dLLM in DPO, reducing the estimation of the dLLM's trajectory probability to a single forward pass.  The two theorems are the two specific ways for it (detailed in section 3).
> 1. In Theorem 3.1, we prove that the total trajectory probability $\pi_\theta(\tau)$ can be unbiased estimated by the sum of expectations of sampled intermediate probabilities.
> 2. In Theorem 3.2, we prove that the probability ratio at each intermediate diffusion step can simplify to the probability ratio of the newly unmasked tokens. By plugging these two theoretical estimations back into the original DPO formula (Equation 6), we derive our final, practical training objective: Equation 12 (detailed in Appendix A.4). Equation 12 is the exact loss function we optimize in our code.
>
> ### **Falling short of the performance of Qwen2.5-Instruct; questions about the practical competitiveness**
> We believe the reviewer's question touches not only on why our specific method does not surpass Qwen2.5-Instruct, but more broadly on why dLLMs generally lag behind strong ARMs.
>
> dLLMs represent a fundamentally different architectural design to ARMs that performs multi-token prediction, not just an ARM with a diffusion add-on. Current dLLM research focuses on converting existing ARMs into dLLMs to improve efficiency through multi-token decoding. During such transition, the information drop is inevitable. The goal of our work—and dLLMs in general—is to provide a favorable trade-off between quality and efficiency. To ensure dLLMs can be practically deployed, their performance must not fall too far behind ARMs. Our method directly addresses this by significantly closing the gap between dLLMs and ARMs, which in turn enhances the dLLM's practical generation efficiency (please refer to the second question of Reviewer DbBn).
>
> ### **The experimental comparison appears somewhat narrow**
> we compared our method against all available dLLMs initialized from Qwen2.5-Instruct to our knowledge, as well as applying other post-training algorithms (like ELBO, VRPO) to the same backbone.
>
> Qwen2.5-Instruct (an ARM) is included in the table not as a direct peer baseline, but as an oracle to show how much of the performance gap our method successfully closes. It would not be meaningful to compare against other unrelated ARMs, as our focus is closing the gap between dLLMs and ARMs while improving the efficiency. Furthermore, we do provide comparisons across a different base architecture (the Qwen3 series) in Table 3. We will clearly highlight the best performance across all models in the revised table.
>
> ### **Details of the algorithm**
> (1)This equivalence holds under the assumption that the token unmasking scheduler is independent of the intermediate states (section 3.1). (2) The relationship is mathematically exact, as shown in Appendix A.2. (3) This trajectory reduction technique can be applied to any policy gradient method that requires a reference policy to regularize the policy gradient.(Appendix A.5)

---

> > ### Author Rebuttal · Reviewer_w4Wk · 2026-04-04
> >
> > Thank you for your response. After reviewing the supplementary material and the code, which I had not examined in detail in my initial assessment, I now have a clearer understanding of the method.

---

### Official Review · Reviewer_TXyb · 2026-03-12

**Soundness:** 3
**Presentation:** 3
**Significance:** 3
**Originality:** 3
**Overall Recommendation:** 4
**Confidence:** 4

**Summary:**

This paper addresses the high computational cost and numerical instability of trajectory probability estimation in preference optimization for diffusion large language models (dLLMs). The authors propose dTRPO, which leverages two theoretical reductions: (i) State Reduction, proving that block-wise sampling of a single timestep yields an unbiased estimate of the full trajectory log-probability, and (ii) Ratio Reduction, showing that schedule-dependent coefficients cancel out in policy ratios, leaving only token-level categorical probabilities. Experiments on 7B dLLMs demonstrate consistent gains across STEM, coding, and instruction-following benchmarks, while matching the training cost of DPO for autoregressive models.

**Compliance With Llm Reviewing Policy:**

Affirmed.

**Final Justification:**

The rebuttal addressed most of my concerns. I believe my rating appropriately reflects the overall quality and contribution of this paper.

**Key Questions For Authors:**

1. Online RL Extension: Appendix A.5 claims the reduction properties generalize to policy gradient methods. Have the authors experimented with online RL scenarios (e.g., GRPO) where trajectories are sampled on-policy, and does the trajectory reduction remain stable under the distribution shift inherent in online training?

2. Baseline Data Consistency: Section 4.2 mentions that some baselines (e.g., Dream) may lead on specific benchmarks due to specialized training data focus, while Section F.4 states all baselines use the same data. Could the authors clarify the exact data mixture for each baseline in Table 2 to ensure the comparison is strictly fair?

3. Empirical Evidence for Numerical Instability: Section 3.3 claims that schedule-dependent coefficients may dominate learned categorical terms with large vocabularies, leading to unstable training. Could the authors provide empirical measurements comparing the magnitude of these coefficients versus $\mu_\theta$ during actual training runs? Furthermore, could you clarify the specific mechanism by which this magnitude mismatch induces instability?

**Limitations:**

See Weaknesses.

**Strengths And Weaknesses:**

### Strengths

1. Solid Theoretical Foundation: The paper establishes a rigorous theoretical framework for dLLM policy optimization. Theorem 3.1 (State Reduction) and Theorem 3.2 (Ratio Reduction) are well-proven in the appendix, providing an unbiased estimator for trajectory probability and mathematically demonstrating the cancellation of schedule-dependent coefficients, which directly addresses the issue of numerical instability.

2. Significant Computational Efficiency: By utilizing block attention masks, the method estimates trajectory probabilities with effectively a single forward pass per block, making the training cost comparable to DPO for autoregressive models while maintaining performance.

3. Comprehensive Experimental Evaluation: The evaluation is extensive and convincing. The method is validated across multiple domains (STEM, coding, instruction-following) with consistent improvements over strong baselines. Detailed ablation studies on scheduling strategies, block sizes, and hyperparameters further substantiate the robustness and effectiveness of the approach.

### Weaknesses

1. Architectural Constraints: The theoretical derivation relies heavily on the absorbing state assumption (Section 3.1), where masked tokens remain masked until unmasked. This limits the method's applicability to non-absorbing diffusion variants that allow token flipping or re-masking during the generation process.

2. Insufficient Sampling in Ablation Studies: The ablation experiments in Figure 3(e) and 3(f) evaluate only 2 and 3 discrete values for the number of samples per block and the top-k ratio, respectively. This sparse sampling makes the results susceptible to random fluctuations and insufficient to establish robust trends or identify optimal operating points.

3. Incomplete Baseline Comparison for Long-Block Models: While Figure 3(c) demonstrates dTRPO's effectiveness on LLaDA, the comparison is limited to the backbone model before and after dTRPO training without benchmarking against other post-training methods (e.g., VRPO, ELBO-SFT).

4. Repeated Empty Appendix Chapters: Sections C and D in the appendix appear to be duplicated headers with no actual content following them in the provided manuscript.

5. Mathematical Indexing Error: In Appendix A.1, the global time index is defined as $sT_B + t$ for block $s \in \{1,...,N_B\}$. However, if $s$ starts from 1, the correct mapping should be $(s-1)T_B + t$ to correctly cover the global time range $[1, T]$.

---

> ### Author Rebuttal · Authors · 2026-03-30
>
> We are glad to see the reviewer found our theoretical foundation solid, noting that it improves stability and efficiency, and recognized extensive experimental validation.
> ### **non-absorbing diffusion variants**
> We appreciate the reviewer's great insight! To clarify, our method can still be applied to non-absorbing diffusion variants as long as the token independence assumption holds.
>
> In the forward  process, we allow the masked token to be revealed as $x\_0$. In the reverse process, we need to modify Equation (4). Defining  $\gamma\_t$ as the probability of re-masking an unmasked token from a previous step, we have
> - $\dfrac{\alpha\_{t-1}-\alpha\_t}{1-\alpha\_t}\,\mu\_\theta(\tau\_{t-1}^{(i)} \mid \boldsymbol{\tau}\_t)$, if $\tau\_t^{(i)}=[M]$ and $\hat{\tau}\_0^{(i)}=\tau\_{t-1}^{(i)}\neq [M]$
> - $\dfrac{1-\alpha\_{t-1}}{1-\alpha\_t}$, if $\tau\_t^{(i)}=[M]$ and $\tau\_{t-1}^{(i)}=[M]$
> - $\gamma\_t$, if $\tau\_t^{(i)}\neq [M]$ and $\tau\_{t-1}^{(i)}=[M]$
> - $1-\gamma\_t$, if $\tau\_t^{(i)}\neq [M]$ and $\tau\_{t-1}^{(i)} = \tau\_t^{(i)}$
> - $0$, otherwise
>
> As in Equation (18) of the paper, the coefficients that do not contain $\mu\_\theta$ will cancel out. Therefore, we can still use the current objective to train the model.
>
> However, we understand that coefficients of non-absorbing variants may rely on the relationships between tokens. This challenge exists in both absorbing and non-absorbing cases. Due to word limit, please see our response to Reviewer DbBn's first question.
> ### **LLaDA with other post-training methods (e.g., VRPO, ELBO-SFT).**
> Due to time constraints, we report these results using the dPad inference protocol with dual cache and a shorter generation length, noted in the brackets.
> | Method | MATH (256) | LCB (1024) | IF (512) |
> | :--- | :--- | :--- | :--- |
> | **LLaDA** | 27.84 |10.29| 55.08|
> | **ELBO-SFT** | 30.71| 7.43| 59.15|
> | **VRPO** | 29.08 | 6.16 |56.93|
> | **dTRPO**|34.88|10.86 | 61|
> ### **online RL scenarios. Does the trajectory reduction remain stable under the distribution shift inherent in online training?**
> We are not sure if we understand the question correctly.
> To our knowledge, there is no distribution shift in data distribution under online training as the data is generated directly by the current policy. In offline training, there is a natural distribution gap between the policy and the external training data. However, offline data can be sourced from stronger models and carefully curated, which help the model to converge faster than online training.
>
> We hope to clarify that the majority of the time is spent on generation in online training.  This leads our trajectory reduction less advantageous.
> Nevertheless, we ran a quick experiment using 100k on-policy data points. We used prompts from the OpenMathReasoning and OpenCodeReasoning datasets, and prompt the model to generate responses ahead of time.
> | Data | MATH | HumanEval+ | IF |
> | :--- | :--- | :--- | :--- |
> | **Baseline (Fast-dLLM-v2)** | 60.26 |  59.1 | 62.11|
> | **Off-policy** |64.3 |63.4 | 65.06|
> | **On-policy** | 62.74 |59.10 | 65.62|
>
> The on-policy data leads to better improvements in general instruction-following (IF) ability during DPO training. However, for rigorous tasks like math and coding, where it is difficult for the model to independently evolve fast toward the correct logic, on-policy training falls short of the performance achieved using high-quality off-policy data.
>
> ### **Data mixture for each baseline in Table 2**
> For publicly released models, we directly evaluated their official checkpoints. For baseline algorithms, we trained them using the exact same backbone model and training data as our method. All the evaluations are under the same protocol.
> It is worth noting that some pre-trained models, such as DREAM, used different proprietary training data, which accounts for better performance in certain benchmarks.
> ### **The magnitude of these coefficients versus $\mu\_\theta$ and the effect of this magnitude mismatch.**
> Consider following Shi et al. (2024) by using a linear schedule $\alpha\_t=1-t$. Here, $t\in[0,1]$ is scaled continuously from $0$ to $1$. One example coefficient evaluates to $\frac{\alpha\_{t-1}-\alpha\_t}{1-\alpha\_t}=\frac{1}{t}$. In practice, if the block size is $N$, this coefficient's value can grow as large as $N$.
> We collected below statistics of $\mu_\theta$ on 100 SmolTalk datapoints:
> - [0.000, 0.001): 14.4%
> - [0.001, 0.010): 13.3%
> - [0.010, 0.050): 15.1%
> - [0.050, 0.800): 31.4%
> - [0.800, 1.000): 25.8%
> As the data shows, when the model struggles with a specific prompt, it assigns a very small probability to the correct token. Generally, when multiplying a very small probability by a very large coefficient, the significant bits of the smaller value ($\mu\_\theta$) are effectively shifted by the large magnitude of the coefficient during floating-point alignment. This loss of precision frequently leads to NaN losses.

---

> > ### Author Rebuttal · Reviewer_TXyb · 2026-04-04
> >
> > Thanks for the detailed rebuttal and for adding the extra results. The clarification on non-absorbing variants and the additional LLaDA comparisons were helpful, and I also appreciate the discussion of the numerical instability issue with concrete statistics. The response addressed several of my main concerns, and I’m glad these clarifications and fixes will be reflected in the final version.

---

### Official Review · Reviewer_DbBn · 2026-03-13

**Soundness:** 4
**Presentation:** 3
**Significance:** 3
**Originality:** 2
**Overall Recommendation:** 5
**Confidence:** 4

**Summary:**

The paper addresses a real bottleneck in post-training diffusion LLMs: computing the probability of a generation trajectory is expensive because, unlike autoregressive models, the probability does not factorize over tokens via causal conditioning. The authors derive two reductions, State Reduction (Theorem 3.1), which shows that sampling one diffusion step per block gives an unbiased estimator of the full trajectory log-probability, and Ratio Reduction (Theorem 3.2), which shows that policy ratios simplify to categorical probabilities over newly unmasked tokens only, with all schedule-dependent coefficients cancelling. Combining these with a DPO-style preference objective yields dTRPO, which requires only 4 forward passes per training example (comparable to standard ARM DPO) and can be implemented with a single forward pass via block attention. Experiments on Fast-dLLM-v2-7B show consistent gains over existing dLLM baselines across instruction-following, STEM, and coding benchmarks.

**Compliance With Llm Reviewing Policy:**

Affirmed.

**Key Questions For Authors:**

1. How does the training objective relate to Theorem 3.1 when a confidence-based scheduler is used?

2. What happens when dTRPO is applied on top of a stronger ARM base (e.g., Qwen3-initialized dLLM)? The current experiments show dTRPO on Fast-dLLM-v2 (Qwen2.5-initialized), but Table 3 hints that the gap between dTRPO and the ARM base may be architecture-dependent.

3. Is the MLP-only training essential for stability, or for performance?

4. What are the mixing ratios of SmolTalk2, math, and code preference data?

**Limitations:**

The authors' impact statement "none which we feel must be specifically highlighted here" is fine for a methods paper, but a few scope limitations deserve acknowledgment. dTRPO inherits all the limitations of the DPO framework, it depends on the quality of preference pairs, is sensitive to the reference model choice, and does not address reward hacking in the preference optimization sense.

**Strengths And Weaknesses:**

## Strengths

1. The theoretical contribution is solid and clean. Theorems 3.1 and 3.2 are not incremental observations, they provide a principled foundation for trajectory probability estimation in dLLMs that existing work had been handling heuristically. The ratio reduction in particular (Theorem 3.2) is a genuinely useful result: the cancellation of schedule-dependent coefficients is non-obvious and has an important practical consequence (training is invariant to schedule choice). The proofs in the appendix are complete and readable.

2. The efficiency comparison in Figure 1 is well-constructed. Positioning dTRPO in the cost-performance space against other dLLM methods is more informative than a standard table, and the argument that dTRPO achieves comparable MATH performance to MATH-specific methods (like SPG and d2) while using general training data and offline training cost is a substantive point.

3. The block attention implementation is elegant. The insight that different blocks can be processed in parallel via a custom attention mask, effectively simulating NB separate partially masked sequences in one forward pass, is a non-trivial engineering contribution that makes the theoretical reduction practically useful. The description in Appendix B is sufficiently detailed.

4. Ablations on the scheduler are informative, and the robustness findings are credible. The result that random-k and top-k perform similarly (Figure 3a), despite top-k violating the independence assumption in Theorem 3.1, suggests the estimator is robust to this approximation in practice. This is an honest and useful finding rather than a post-hoc rationalization.


## Weaknesses

1. The connection between the theoretical formulation and the actual training objective deserves more care. Theorem 3.1 states an unbiased estimator for the log-trajectory-probability under the assumption that block steps are sampled uniformly. But in practice, the paper uses a confidence-based top-k scheduler to select which tokens to unmask, which is not the same process as sampling a uniform timestep and then unmasking uniformly at random.


2. The inference speed claim deserves more scrutiny. Figure 3(d) reports that dTRPO achieves 1.9 speedup in TPS over Qwen2.5-7B-Instruct. But this comparison is between a dLLM (which generates in blocks in parallel) and an ARM (which generates autoregressively). This maybe a property of the diffusion architecture, not of dTRPO the training method.

---

> ### Author Rebuttal · Authors · 2026-03-30
>
> We are glad to see that the reviewer recognized our theoretical development of the problem as a "principled foundation" rather than an "incremental observation." Here are our responses:
>
> ### **What if uniform sample is violated?**
>
> Overall, we believe this is  not a trivial problem but an interesting direction with significant scope for future research. Below is the starting point.
>
> Let $\boldsymbol{\tau}\_t \in \mathbb{R}^N$ be the token sequence at time $t$. The forward process should then take the following vector form:
> $$q(\boldsymbol{\tau}\_t | \boldsymbol{\tau}\_{t-1}) = (1 - \beta\_t(\boldsymbol{\tau}\_{t-1})) \mathbf{1}\_{\{\boldsymbol{\tau}\_{t-1} = \boldsymbol{\tau}\_t\}} + \beta\_t(\boldsymbol{\tau}\_{t-1}) \mathbf{1}\_{\{\boldsymbol{\tau}\_{t-1} \neq \boldsymbol{\tau}\_t\}}$$
> where $\beta\_t(\cdot): \mathbb{R}^N \to \mathbb{R}^N$ is the transition probability function, and $\beta\_t(\boldsymbol{\tau}\_{t-1}) \in \mathbb{R}^N$ is the element-wise probability vector of each token in the sequence $\boldsymbol{\tau}\_{t-1}$ being unmasked at time $t$. $\mathbf{1}\_{\{\boldsymbol{\tau}\_{t-1} = \boldsymbol{\tau}\_t\}}$ is a vector of length $N$, where an element is 1 if the corresponding token is the same between $\boldsymbol{\tau}\_{t-1}$ and $\boldsymbol{\tau}\_t$, and 0 otherwise. Here the multiplication is element-wise.
> To approximate the posterior, we use the following form to parameterize the transition probability function:
> $$\beta\_t(\boldsymbol{\tau}\_t) = \mathbf{1}\_{\{h(\mu(\boldsymbol{\tau}\_t | \boldsymbol{\tau}\_{t-1}))\}}$$
> where $\mathbf{1}$ is applied element-wise, and we simply denote $h$ as any confidence based scheduler. With this parameterization, the remaining formulation can be written in a similar format. The main difference lies in Theorem 3.2, where we hope the coefficients can be canceled between the policy and the reference. Since the output policy probability $\mu\_\theta(\boldsymbol{\tau})$ and the output reference probability $\mu\_\text{ref}(\boldsymbol{\tau})$ differ slightly, the coefficients $\beta\_t$ are not the same for the same trajectory, thus cannot be canceled exactly.
> Future research could provide insights on how to make up for the difference between the policy-parameterized and reference-parameterized transition probability functions $\beta\_t$.
>
> The complexity of this exact problem is why we chose to directly validate the effectiveness of the top_k scheduler through experiments in the current manuscript.
>
> ### **The inference speed claim deserves more scrutiny.**
>
> We provide a more detailed comparison between Qwen2.5, Fast-dLLM-v2, and our method below:
>
> | Model | GSM8K TPS | GSM8K Accuracy | Arena-Hard TPS | Arena-Hard Score |
> | :--- | :--- | :--- | :--- | :--- |
> | **Qwen2.5-7B-Instruct** | 38.9 | 87.87 | 16.20 | 10.43 |
> | **Fast-dLLM-v2** | 38.84 | 82.34 | 19.55 | 6.79 |
> | **dTRPO** | 38.80 | 85.97 | 29.87 | 7.41 |
>
> While inference speed is inherently a property of the diffusion architecture, it is also closely tied to generation quality. As seen in the GSM8K results, the TPS is similar across all models because they only need to generate short responses. However, on the more difficult Arena-Hard benchmark, dLLMs are generally faster than ARMs, and compared to Fast-dLLM-v2, our method achieves higher TPS.
>
> ### **What happens when dTRPO is applied on top of a stronger ARM base?**
>
> We appreciate the reviewer's keen observation! We want to clarify that our choice of backbone was primarily driven by training efficiency considerations, as one of our main goals is to efficiently scale up dLLM post-training.
>
> We chose the diffusion base model Fast-dLLM-v2 because it has better infrastructure support, allowing large-scale training to run much faster. Existing post-training work on the diffusion bases trained from Qwen3, with worse infrastructure support, has mostly been conducted on a small scale, focusing specifically on math and reasoning (~14k data). In contrast, our method uses 500k data points to provide a strong general starting point for diverse downstream tasks. Completing large scale training on other backbones still requires substantial community effort in base training and infrastructure development.
>
> ### **Is the MLP-only training essential for stability, or for performance?**
> This design choice is primarily for stability, as DPO training generally requires only minimal changes to the original model to avoid degrading base capabilities, as shown in Figure 5(a).
>
> ### **What are the mixing ratios of SmolTalk2, math, and code preference data?**
> We simply mix them together evenly. Although the SmolTalk2 dataset is much larger than the other two, its data is less diverse and mostly consists of linguistic tasks requiring the model to write specific types of sentences.
> | Data Source | MATH | LCB | IF |
> | :--- | :--- | :--- | :--- |
> | **SmolTalk2 only** | 63.18 | 13.36 |67.10|
> | **SmolTalk2 + Math + Code** | 64.3 | 15.17| 65.06 |

---

> > ### Author Rebuttal · Reviewer_DbBn · 2026-04-02
> >
> > The Authors have fully solved all my questions. This is a technically solid paper which should be accepted.

---

### Decision · Program_Chairs · 2026-04-30

**Decision:**

Accept (regular)

**Comment:**

After carefully considering the four reviews, I recommend a “accept” for this paper. The reviewers unanimously recognize solid theoretical contributions, particularly the state and ratio reductions that enable efficient and stable trajectory probability estimation for diffusion LLMs, which addresses a key bottleneck in aligning these models. The empirical results on 7B-scale models show consistent improvements over baselines, and the block attention implementation is an elegant engineering contribution. However, several concerns temper enthusiasm. The evaluation is limited to a single model family and scale (7B dLLMs), leaving generalizability to other architectures and sizes unproven. Some reviewers note that the performance still lags behind strong autoregressive baselines like Qwen2.5 Instruct, raising questions about practical competitiveness. Additionally, the theoretical derivations assume an absorbing-state diffusion process, limiting applicability to other diffusion variants, and the connection between the theory and the practical confidence-based scheduler is not fully justified. While the paper is technically sound and advances a niche but important sub-area, these limitations in evaluation breadth and architectural generality prevent a stronger recommendation.